# Genome binning of viral entities from bulk metagenomics data

Joachim Johansen [1,2], Damian R. Plichta [2], Jakob Nybo Nissen[1,3], Marie Louise Jespersen[1,4], Shiraz A. Shah [5], Ling Deng[6], Jakob Stokholm [5,6], Hans Bisgaard [5], Dennis Sandris Nielsen [6], Søren J. Sørensen [7] & Simon Rasmussen [1✉]

Despite the accelerating number of uncultivated virus sequences discovered in metagenomics and their apparent importance for health and disease, the human gut virome and its interactions with bacteria in the gastrointestinal tract are not well understood. This is partly due to a paucity of whole-virome datasets and limitations in current approaches for identifying viral sequences in metagenomics data. Here, combining a deep-learning based metagenomics binning algorithm with paired metagenome and metavirome datasets, we develop Phages from Metagenomics Binning (PHAMB), an approach that allows the binning of thousands of viral genomes directly from bulk metagenomics data, while simultaneously enabling clustering of viral genomes into accurate taxonomic viral populations. When applied on the Human Microbiome Project 2 (HMP2) dataset, PHAMB recovered 6,077 high-quality genomes from 1,024 viral populations, and identified viral-microbial host interactions. PHAMB can be advantageously applied to existing and future metagenomes to illuminate viral ecological dynamics with other microbiome constituents.

[1] Novo Nordisk Foundation Center for Protein Research, Faculty of Health and Medical Sciences, University of Copenhagen, Copenhagen, Denmark. [2] Infectious Disease and Microbiome Program, Broad Institute of MIT and Harvard, Cambridge, MA, USA. [3] Statens Serum Institut, Viral & Microbial Special diagnostics, Copenhagen, Denmark. [4] National Food Institute, Technical University of Denmark, Kongens Lyngby, Denmark. [5] Copenhagen Prospective Studies on Asthma in Childhood (COPSAC), Herlev and Gentofte Hospital, University of Copenhagen, Copenhagen, Denmark. [6] Section of Food Microbiology and Fermentation, Department of Food Science, Faculty of Science, University of Copenhagen, Copenhagen, Denmark. [7] Section of Microbiology, Department of Biology, University of Copenhagen, Copenhagen, Denmark. ✉email: simon.rasmussen@cpr.ku.dk

The human gut microbiota is tightly connected to human health through its massive biological ecosystem of bacteria, fungi, and viruses. This ecosystem has been profoundly investigated for discoveries that can lead to diagnostics and treatments of gastrointestinal diseases such as inflammatory bowel disease (IBD) and colon cancer as well as type 2 diabetes (T2D)[1–3]. In IBD, multiple studies have compiled a list of keystone bacterial species undergoing microbial shifts between inflamed and non-inflamed tissue sites[4,5] and there are strong indications that the gut virome plays a role in disease aetiology[6–8]. Now, the influence of bacteria-infecting viruses, known as bacteriophages, are increasingly studied and their role in controlling bacterial community dynamics in the context of gastrointestinal pathologies is slowly being unravelled[9]. Several studies have presented evidence of temperate *Caudovirales* viruses increasing in Crohn's disease (CD) and ulcerative colitis (UC) patients[6,8,10,11]. However, it has been left unanswered if this phage expansion was due to alterations in host-bacterial abundance, thus viral-host dynamics remains another unexplored facet of the gut virome in diseases such as IBD[12].

Today, the virome is studied through metagenomics where high-throughput sequencing is computationally processed to construct genomes of uncultivated viruses de novo. Viral assembly is a notoriously difficult computational task and is known to produce fragmented assemblies and chimeric contigs[13] especially for rare viruses with low and uneven sequence coverage[14,15]. For better viral assemblies, metaviromes are prepared with extra size-filtration to increase the concentration of viral particles[16,17]. However, identification of viruses without enrichment from bulk metagenomics, is increasingly utilised and overcomes the size-filtration step biases while enabling identification of primarily temperate but also lytic viruses[18]. Currently, several approaches for identifying viral sequences in metagenomics data exist and have helped in supersizing viral databases of uncultivated viral genomes (UViGs) over the last few years[19–21]. These tools are often based on sequence similarity[22], sequence composition[23–28], and identification of viral proteins or the lack of cellular ones[27,28]. A common denominator for these tools is their per-contig/sequence virus evaluation approach that is not optimal for addressing fragmented multi-contig virus assemblies.

Therefore, we developed a framework (PHAMB) based on contig binning to discover viral genome bins directly from bulk metagenomics data (MGX). For this, we utilised a recently developed deep-learning algorithm for metagenomic binning (VAMB)[29] that is based on binning the entire dataset of assembled contigs. Altogether, we reconstructed 2676 viral populations from bulk metagenomes corresponding up to 36% of the paired metavirome dataset (MVX), based on two independent datasets with paired MGX and MVX. A key development in our method is a classifier that can classify non-phage bins from any dataset with very high accuracy (93–99%) compared to existing virus prediction tools such as DeepVirFinder (69–74%)[25], Virsorter2 (30–84%)[30] and viralVerify (86–98%)[31]. Our approach enables identification and reconstruction of viral genomes directly from metagenomics data at an unprecedented scale with up to 6077 viral populations with at least one High-Quality (HQ) genome by MIUViG standards[18] in a single dataset. In addition, we show an increase of up to 210% of HQ viral genomes extracted by combining contigs into viral bins. Using this method to extract viruses from the microbial metagenomes of the HMP2 cohort we were able to delineate both viral and bacterial community structures. This allowed us to investigate viral population dynamics in tandem with predicted microbial hosts for instance identifying 123 and 230 viral populations infecting *Faecalibacterium* and *Bacteroides* genomes, respectively.

## Results

### A framework to bin and assemble viral populations from metagenomics data

To generate the metagenomics bins we used VAMB that has the advantage of both binning microbial genomes, and grouping bins across samples into subspecies or conspecific clusters. This has proven useful for the investigation of bacterial and archaeal microbiomes, but the approach has even more potential within viromics as viruses are much less conserved, more diverse, and harder to identify without universal genetic markers such as those found in bacterial organisms[32]. Clusters of conspecific viral genomes would enable straightforward identification and tracking of populations across a cohort of samples (Fig. 1a). To develop our framework we used two Illumina shotgun sequencing-based datasets with paired metagenome and metavirome available. The Copenhagen Prospective Studies on Asthma in Childhood 2010 (COPSAC) dataset consisted of 662 paired samples (refs. [33,34]) and the Diabimmune dataset contained 112 paired samples[35]. Each of the two datasets included a list of curated viral species, 10,021 and 328 respectively, that we used here as our gold standard for training and testing our tool. Compared to COPSAC, Diabimmune metaviromes had low viral enrichment (Supplementary Fig. 1), we, therefore, used the average amino acid identity (AAI) model of CheckV[28] to stratify the genomes of the metaviromes into quality tiers ranging from Complete, High-Quality (HQ), Medium-Quality (MQ), Low-Quality (LQ) and Non Determined (ND) to establish a comparable viral truth.

### Viral binning is more powerful compared to single-contig approaches

The output of binning metagenomic samples can be hundreds of thousands of bins and we therefore first developed a Random Forest (RF) model to distinguish viral-like from bacterial-like genome bins. The RF model takes advantage of the cluster information from binning and aggregates information across sample-specific bins to form subspecies clusters. Here, we found that the RF model was able to separate bacterial and viral clusters very effectively with an Area Under the Curve (AUC) of 0.99 and a Matthews Correlation Coefficient (MCC) of 0.91 on the validation set (Fig. 1b and Supplementary Table 1). Compared to single-contig-evaluation methods, the RF model was superior as other methods achieved an AUC of up to 0.86 and MCC up to 0.16. This difference in performance is likely explained by the RF model evaluating on bin-level where one sequence with a low viral score does not lead to a misprediction of the whole bin. For instance, we achieved an increase of 200 (190%) and 771 (95%) HQ bins recovered for the Diabimmune and COPSAC datasets compared to using single-contig-evaluation according to CheckV (Fig. 1c, d). Based on the single-contig CheckV evaluations, we found that 97.7 and 95.3% of HQ contigs were binned into HQ bins in COPSAC and Diabmmune, respectively. This means that a small percentage of the HQ contigs, up to 2.3 and 4.7%, are lost in the binning process at the expense of a net increase in genome recovery but can be recovered by parallel single-contig evaluations. Finally, we observed a significantly greater number of viral hallmark genes per virus when using viral bins in both datasets (T-test, two-sided, $t = 16.85$, $P < 0.0005$), while the length and viral fraction were largely comparable (Supplementary Fig. 2).

### High viral binning performance on simulated viromes

We then investigated the viral binning performance of VAMB and the prediction performance with simulated datasets including two pure viral and one mixed dataset containing bacteria, plasmids and viruses. The two pure viral datasets comprised 80 crAss-like viruses and 50 small-genome (<6000 bp) randomly sampled from the MGV database[20]. To establish the mixed dataset, the crAss-

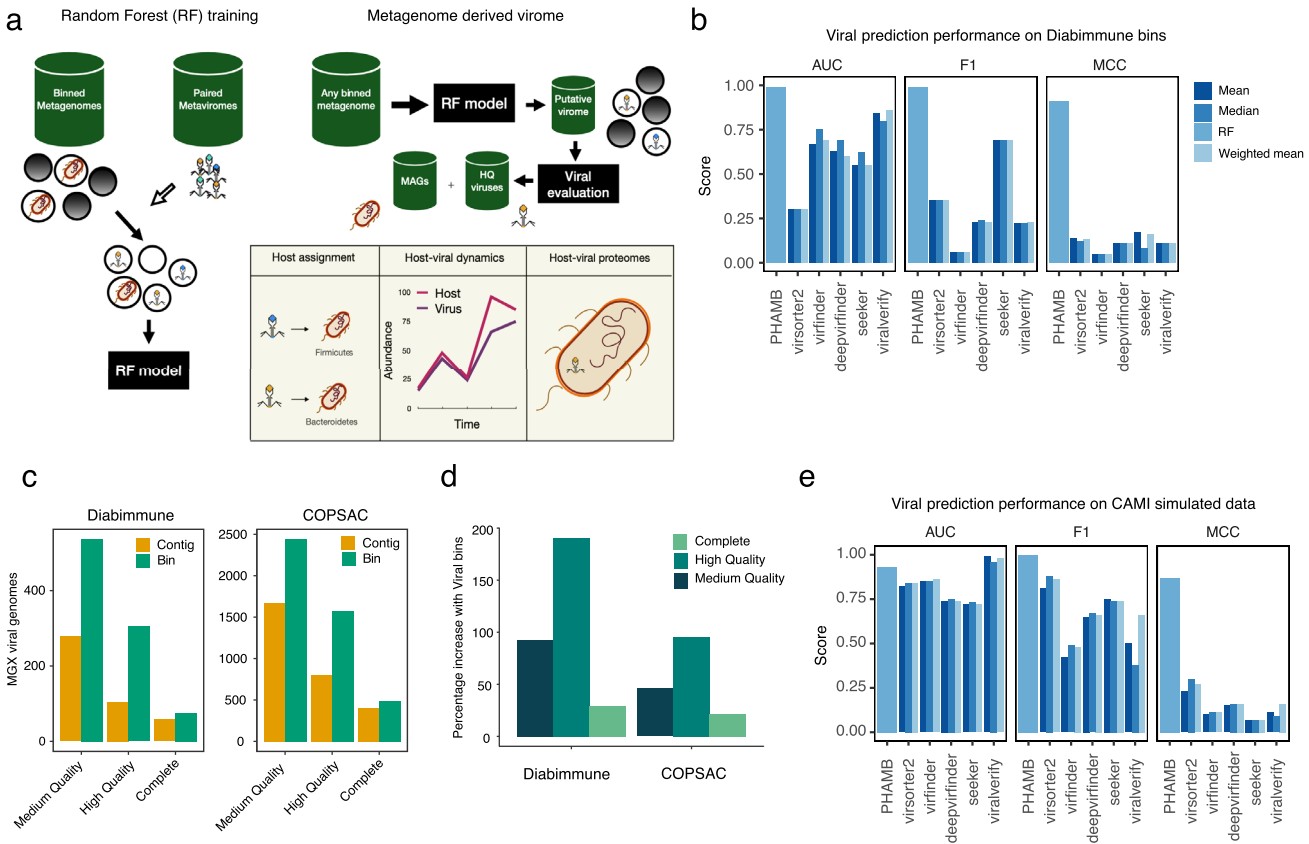

**Fig. 1 A framework to bin and assemble viral populations from metagenomics data. a** Illustration of workflow to explore viruses from binned metagenomes. First, the RF model was trained on binned metagenomes; bacterial bins were identified using reference database tools and viruses were identified using assembled viruses from paired metaviromes. Viral and bacterial labelled bins were used as input for training and evaluating the RF model. Bins from any metagenome such as human gut, soil or marine can be parsed through the RF model to extract a space of putative viral bins that are further validated for HQ viruses using dedicated tools like CheckV. Binned MAGs and viruses can then be associated in a host assignment step. Host-viral dynamics can be explored in longitudinal datasets to establish temperate phages and the contribution of viruses to Host pangenomes **b** AUC, F1-score and Matthews correlation were calculated for prediction results on viral bins from Diabimmune. These performance scores were calculated based on probability scores from the trained RF model and summarised viral bin-scores of various viral prediction tools. For all tools except the RF model, genomes were labelled viral if the summarised viral score across all contigs, calculated either as a mean, median or contig-length weighted mean passed a threshold. The following thresholds used were 7, 0.5, 0.9, 0.9, 0.9 for viralVerify, Seeker, Virsorter2, Virfinder and DeepVirfinder, respectively. **c** The number of viral genomes recovered from bulk metagenomes, counted at three different levels of completeness in Diabimmune or COPSAC cohorts, evaluated as either single-contigs or viral bins from bulk metagenomes. Evaluation of genome completeness was determined using CheckV here shown for MQ ≥ 50%, HQ ≥ 90%, Complete = Closed genomes based on direct terminal repeats (DTR) or inverted terminal repeats. **d** The percentage-increase of viral genomes found in Diabimmune or COPSAC cohorts using our approach relative to single-contig evaluation. The increase is coloured at three different levels of completeness determined using CheckV, corresponding to the ones used in (**c**). **e** Similar to (**b**) prediction performance scores were calculated for the trained RF model and various viral predictors but on prediction results of CAMI simulated viral genomes from the mixed genome set including bacteria, viruses and plasmids. MAGs metagenome-assembled genomes, HQ high-quality, MQ medium-quality and AUC area under curve.

like and small-genome datasets were combined with an additional 150 random virus genomes, 8 bacterial genome isolates and 20 plasmids (see methods). On the mixed dataset, VAMB outperformed MetaBAT2 on bins with high >0.9 recall and >0.9 precision with a total of 144 vs 134 bins, corresponding to just above 50% (144/280) of all simulated virus genomes (Supplementary Fig. 3a). Furthermore, we found that VAMB binned increasingly a higher number of bins at lower recall (>0.5) and increasing precision levels. Regarding plasmids, both tools were comparable and binned up to 10/20 plasmids with >0.5 recall and >0.95 precision (Supplementary Fig. 3b). Next, we addressed how binning performance could be influenced by virus genome size and highly-similar viruses. For this we sampled smaller virus genomes (<6000 bp, $n = 50$) and viruses of the same family (crAss-like, $n = 80$). A total of 48/50 and 70/80 genomes were binned with >0.99 recall and >0.99 precision for the small-virus and same family-virus set, respectively (Supplementary Fig. 4ab).

The ease of binning small viruses was confirmed in the mixed dataset where VAMB captured the majority of small viruses with high recall and precision (F1 > 0.9) (Supplementary Fig. 4c), indicating that genome size was less confounding to binning performance. Finally, to further validate the RF model, we compared the performance in predicting if a bin was viral or bacterial to single-contig viral predictors (Fig. 1e). Using the mixed simulated dataset the single-contig methods displayed much lower discriminatory performance compared to the RF model. For instance, multiple single-contig viral predictors with a high AUC (up to 0.98) displayed low MCC scores meaning that the prediction was not very accurate at the given threshold (Fig. 1e and Supplementary Figs. 5, 6). We then tried to optimise the decision threshold for each of the single-contig viral predictors (Supplementary Figs. 5, 6) which improved the MCC slightly. For instance, viralVerify achieved an AUC of 0.98 on the simulated data, showing that it was effective in separating bacterial and viral

genomes, however with an overlap in the bacterial and viral score distributions. Therefore, even with an optimised threshold, viralVerify displayed an MCC of 0.39. In contrast, the RF model displayed both high AUC (0.93) and MCC (0.87). Thus, we found the RF model, followed by viralVerify, to be the best-suited method on bin-level in mixed-organism assembly datasets. While the RF model predicts plasmids incorrectly as viral, we found that the downstream use of CheckV helped in making a final confident evaluation as plasmid bins contain multiple bacterial-origin genes and are typically classified as 'NA' or picked up by the less precise HMM-model (Supplementary Fig. 7).

**Binning the metagenome identifies viral genomes not identified from the metavirome.** When applying our method of binning with VAMB and the RF model we obtained 4,480 and 916 viral bins with an MQ or HQ representative bin across the COPSAC and Diabimmune datasets, respectively. We then considered all VAMB clusters as 'viral populations' and thus obtained 2428 and 534 viral populations with at least 1 MQ or better viral bin. After comparing the viral populations obtained from the metagenomics datasets to the respective metaviromes we recovered 17–36% of HQ viruses (corresponding to 527 and 2676 metaviromic viral populations) established in the metaviromes on species (ANI > 95) level and 9–28% on strain (ANI > 97) level (Fig. 2a). The fraction of viruses in the metavirome recovered in the metagenome was considerably higher than more recent estimates[36], which estimated 8.5–10%. This was interesting since the deeply sequenced metavirome may capture multiple low abundant viruses typically not found in metagenomes. Additionally, we found that 46–69% of the HQ metagenome viral populations, corresponding to 124 in Diabimmune and 839 viral populations in COPSAC, were not found in the metavirome, suggesting that a significant part of the virome may be lost during viral enrichment or not represented in induced forms as they are integrated prophages (Fig. 2b). However, we also found that 65–83% of the HQ viral populations in the metavirome were not found in the metagenome data (total 197 in Diabimmune and 2589 in COPSAC) suggesting the reverse to be true as well. For a subset of the viruses found in the COPSAC bulk and metavirome, we estimated higher mean completeness with viral bins (T-test, two-sided, $T = 34.02$, CI = 24.4;27.4, $P = 2.2e-16$) (Fig. 2c). Altogether we found that a great proportion of the gut viral populations can be reconstructed from the metagenomics data and retrieved with even higher completeness compared to the metavirome counterparts.

**Viral bins have low contamination.** Lastly, we wanted to investigate the occurrence of technically 'misbinned' and contaminating contigs that could inflate viral genome size and influence evaluation and downstream analyses. Based on the viral bins ($n = 1705$) that were highly similar to metavirome viruses in the COPSAC dataset (see Methods), we found in 91.4% of all cases, each bin contained no unrelated contigs (Fig. 2d). Considering only multi-contig bins ($n = 570$) we calculated an average bin-purity of 97.4% in base pairs (median 100%), meaning that on average 2.55% of the genome was not aligning to the corresponding MVX virus. This indicates contamination or, alternatively, a more complete virus in the bulk metagenomic dataset. We further investigated the extent of contamination based on simulated data where 87.6% of the viral bins had a precision of 1 (Supplementary Fig. 8a). For multi-contig bins, we calculated an average bin-purity of 94.5% (median 100%) supporting the results on real data that the majority of bins have low contamination. In summary, our combined binning and machine learning approach improves identification and recovery of viral genomes from metagenomics data and outlines the possibility of binning both fragmented and

complete viruses directly from human gut microbiome samples with low degrees of contamination.

**Reconstructing the virome of the HMP2 IBD gut metagenomics cohort.** We then applied our method to the HMP2 IBD cohort consisting of 27 healthy controls, 65 CD, and 38 UC patients[37]. These samples were gathered in a longitudinal approach and consisted of between 1–26 samples per patient. Importantly, no characterised metaviromics data is available from this cohort and using our approach we were able to identify bacterial and viral populations in the cohort and explore their dynamics in IBD using only metagenomics data. From the cohort, we recovered 577 Complete, 6077 HQ, 9704 MQ (Fig. 3a) and 122,107 LQ viral bins corresponding to 263 Complete, 1024 HQ, 2238 MQ and 44,017 LQ viral populations. We also observed an increase in genome completeness for larger viruses/jumbo viruses with a genome size >200 kbp[38] compared to a single-contig evaluation (Supplementary Fig. 9). Across all the datasets we observed 54 binned putative jumbo viruses (Supplementary Data 1). In addition, we observed that similar viral length distributions for viruses recovered as a single-contig and as viral bins, both correlated with CheckV quality tiers (Fig. 3b).

**Viral population taxonomy is highly consistent.** We then investigated the taxonomic consistency of our viral populations and found this to be very high as the median intra-cluster Average Nucleotide Identity (ANI) for MQ to Complete viral clusters was 97.3–99.3% (Supplementary Fig. 11). Even in clusters with over 100 sample-specific viral bins the intra-cluster median ANI was consistently high (median = 97.1–98.5%) (Fig. 3c). Inter-cluster ANI was much lower in the 91.7–92.8% range closer to the genus level. Therefore, our approach was able to identify and cluster near strain-level viral genomes across samples. For example, in the HMP2 dataset, we identified 50 different viral populations for a total of 916 MQ or better crAss-like viral bins. Here, viral population 653 corresponded to the prototypic crassphage[39] and accounted for 253 of the 916 crAss-like genomes discovered in the HMP2 dataset. We then used all of these 916 bins to generate a phylogenetic tree based on the large terminase subunit (TerL) and found the highly consistent placement of the viral genomes according to their binned viral population (Fig. 3d and Supplementary Fig. 12). Viral population 653 formed one monophyletic clade except for one bin while all the other crAss-like clusters were monophyletic. The division of the crAss-like genomes into the binned clusters therefore likely represents actual viral diversity. Taken together, this shows that our reference-free binning produces taxonomically accurate viral clusters, thus aggregating highly similar viral genomes across samples.

**The metagenomic virome is personal and highly stable in healthy subjects.** Several metavirome studies have reported the presence of stable, prevalent and abundant viruses in the human gut[7,40]. We found that the gut virome in the HMP2 cohort[37] was highly personal and stable over time in nonIBD subjects, which was reflected by the lower Bray–Curtis dissimilarity between samples from nonIBD subjects compared to UC (T-test, two-sided $P = 0.017$, $t = −2.47$, CI = −0.01;−0.13) and CD subjects (T-test, two-sided, $P = 0.023$, $t = −2.3$, CI = −0.12;−0.01) (Fig. 4a, b). In addition, the dysbiotic samples, as defined by Price et al. (2019)[37], could be clearly separated with a principal component analysis (PCoA), where the virome explained 4.2 and 3.4% of the variation (Fig. 4c). This was confirmed with a PERMANOVA test on viral ($P < 10 − 3$, $R^2 = 1.6\%$, $F = 9.51$, permutations = 999) and bacterial abundance profiles ($P < 10 − 3$, $R^2 = 3.0\%$, $F = 11.97$) and shows dysbiosis affecting both the virome and bacteriome. Alpha-diversity metrics supported this as

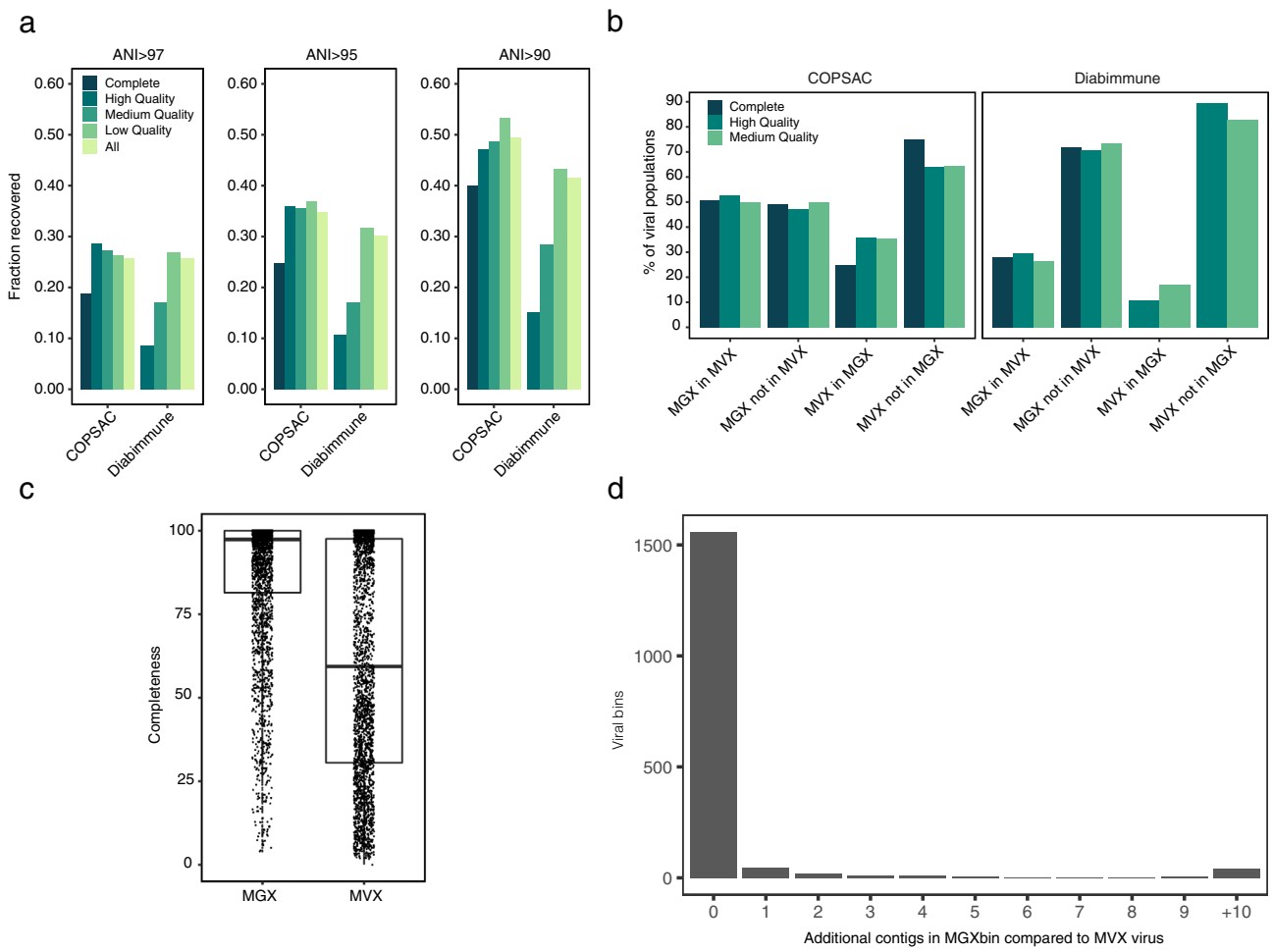

**Fig. 2 Binning the metagenome identifies viral genomes not identified from the metavirome. a** The fraction of metavirome viruses in COPSAC and Diabimmune coloured at different levels of completeness or all together determined with CheckV, identified in VAMB bins from bulk metagenomics of the same cohorts. We defined a metavirome virus to be recovered if the aligned fraction was at least 75% and ANI was >90, >95 or >97.5 to a VAMB bin based on FastANI. **b** The percentage of viral populations, at different levels of completeness determined with CheckV, identified in both metaviromes (MVX) and bulk metagenomics (MGX) or unique to either dataset. Shared populations are identified with a minimum sequence coverage of 75% and ANI above 95%. (1) MGX in MVX: % of Viral populations found in MGX also found in MVX. (2) MGX not in MVX: % of Viral populations unique to MGX i.e. not found in MVX. (3) MVX in MGX: % of Viral populations found in MVX are also found in MGX. (4) MVX not in MGX: % of Viral populations unique to MVX i.e. not found in MGX. **c** Viral genome completeness estimated for $n = 2646$ viruses found both in metaviromes and bulk metagenomics sharing the same nearest reference in the CheckV database. **d** The number of contigs in viral bins from bulk metagenomics that do not align to the closest viral reference in the metavirome. In the majority of viral bins, all contigs align to the nearest reference. ANI average nucleotide identity.

Shannon-Diversity (SD) was higher in nonIBD subjects compared to both UC and CD ($T$-test, two-sided, $P = 0.000155$, $t = -3.79$ and $P = 7.9e-09$, $t = -5.81$) while dysbiosis affected every patient group resulting in a significantly reduced SD. In accordance, viral richness was lower in UC (two-sided $T$-test, $P = 1.44e-15$, $t = -8.09$, $CI = -12.40; -19.80$) and CD (two-sided $T$-test, $P = <2e-16$, $t = -9.39$, $CI = -12.91; -19.50$) patients and further exaggerated in dysbiotic samples (Fig. 4d, e). These viral alpha-diversity trends were also observed in the bacteriome, suggesting that the viruses follow the expansion or depletion of their bacterial host during dysbiosis (Supplementary Fig. 14). Indeed, we identified 250 likely temperate viruses out of 348 differentially abundant viruses that expanded with increasing dysbiosis (linear-mixed-effect model, adj. $P < 0.005$, FDR-corrected). This observation acknowledges earlier results showing an increase in temperate viruses in UC and CD[6,10]. Further analysis on the longitudinal abundance profiles of virus and predicted bacterial host reaffirmed the synchronised expansion theory (Supplementary Fig. 15).

**Viral–host interactions can be explored from viral populations and MAGs**. A unique feature of performing the analysis on metagenomics data is that both the bacterial and viral populations are binned simultaneously. Therefore, we were able to estimate the abundance of both the viral and bacterial compartments of the microbiome and explore the viral host range in silico using the MAGs. In total from the HMP2 dataset, we obtained 3130 and 3819 Near-Complete (NC) and Medium-Quality (MQ) MAGs[41]. Based on MAG-derived CRISPR spacers we found spacer hits to 464 (45.3%) to viral populations with at least one HQ representative. To further expand our viral-host prediction we conducted an all-vs-all alignment search between the MAGs and viral populations for prophage signatures. Then by combining the CRISPR spacer and prophage search we connected 93.6, 74.4, 82.5 and 65.0% of MAGs from *Bacteroidetes, Firmicutes, Actinobacteria,* and *Proteobacteria phylum,* respectively, with at least one virus (Supplementary Fig. 16). We estimated host-prediction purities to be 94.5 and 75.6% on species rank for the CRISPR spacer and prophage signature (Supplementary

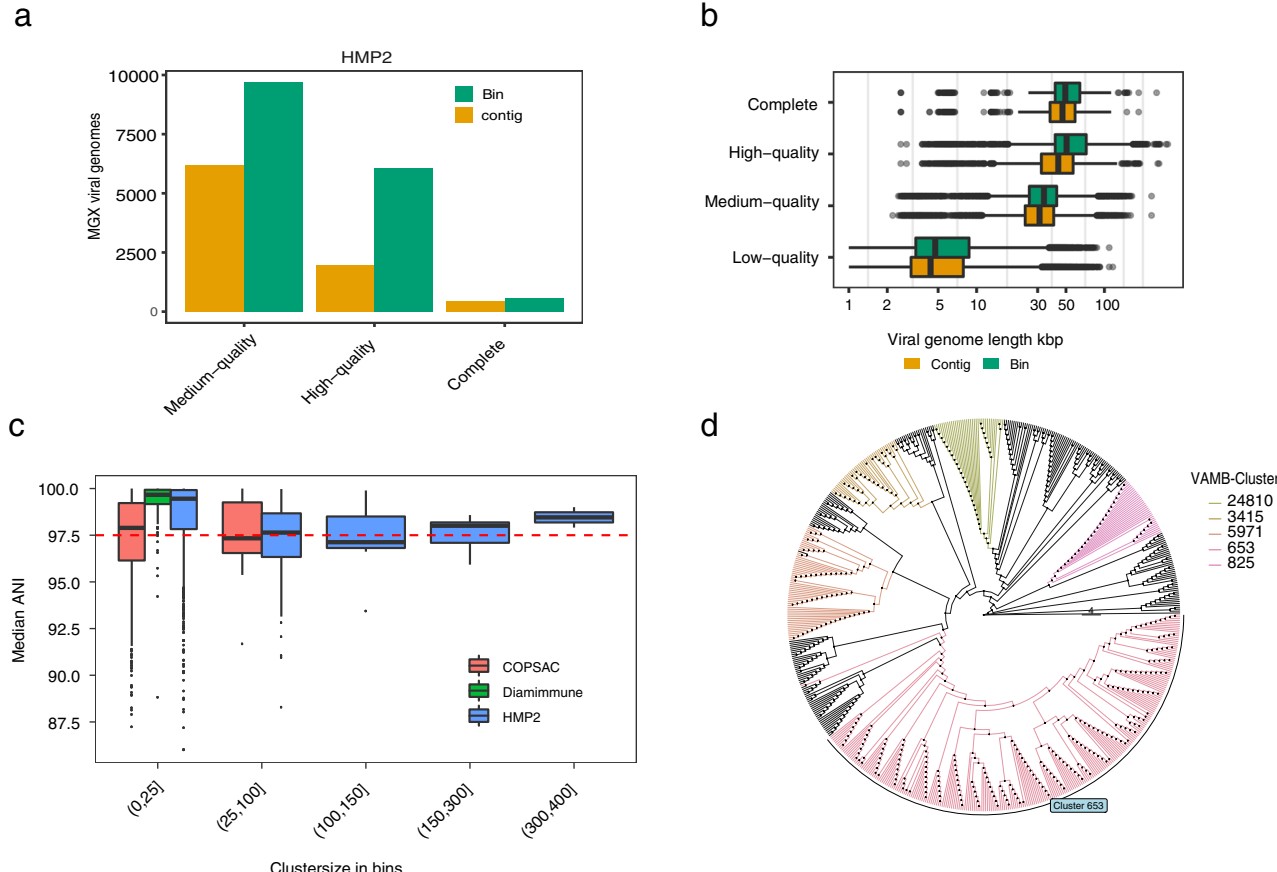

**Fig. 3 Reconstructing the virome of a human gut metagenomics cohort. a** The number of viral genomes with three different levels of completeness in HMP2, evaluated as either single-contigs or viral bins from bulk metagenomes. Evaluation of genome completeness was determined using CheckV here shown for medium-quality ≥50% (MQ), high-quality ≥90% (HQ), Complete = closed genomes based on direct terminal repeats or inverted terminal repeats. **b** The sequence length distribution in kbp of viral genomes at four different levels of completeness in HMP2, evaluated as either single-contigs (n = 215,009) or viral bins (n = 138,367) from bulk metagenomes. Shown for low-quality (LQ) <50%, MQ, HQ and Complete. **c** Median ANI based on pairwise ANI genome measurements between bins within the same VAMB cluster. Median ANI is consistently above 97.5 in small VAMB clusters with 0–25 bins and in larger VAMB clusters with 300–400 bins. **d** Cladogram of an unrooted phylogenetic tree with crAss-like bins based on the large terminase subunit protein (TerL). Five different VAMB clusters have been coloured and illustrate high monophyletic relationships. The phylogenetic tree was constructed using IQtree using the substitution model VT + F + G4. ANI average nucleotide identity %, DTR direct terminal repeats, ITR inverted terminal repeats, Kbp kilobase pairs.

Fig. 17B). Therefore, we confirmed that most gut phages have a primarily narrow host range[42]. MAGs belonging to the genera *Faecalibacterium* and *Bacteroides* seemed to be viral hotspots since 99.7 to 98.7% could be associated with a HQ viral bin, corresponding to 123 and 230 distinct viral populations, respectively (Fig. 5a). For instance, in abundant commensals like *Bacteroides vulgatus* (cluster 216) we observed consistent prophage signals over time for multiple viruses across several samples (Fig. 5b). Interestingly, because the host range of crAss phages are not well understood we investigated CRISPR spacer hits to the MAGs in our databases. Even though we could host-annotate an overall of 45.3% of all HQ viral populations to a MAG, only 74 of the 916 crAss-like bins could be associated with any of the 3306 *Bacteroidetes* bins in our dataset using CRISPR spacers. This was despite having assembled CRISPR arrays (with confidently predicted subtypes) for 998/3306 (~30%) of the *Bacteroidetes* bins. When we performed a similar search to a comprehensive CRISPR spacer database[43] of 580,383 bacterial genomes we could annotate 512 of the 916 crAss-like bins to Bacteroidetes bacteria. These findings suggest that crAss-like phages are not frequently targeted by CRISPR spacers extracted from *Bacteroides* CRISPR-Cas systems within the same environment.

**The binned viral populations are enriched in proteins found in temperate phages.** Another topic of interest was viral-host complementarity, in particular, what functions bacteriophages could provide to the host and how the viral proteome differs with respect to host taxonomy. Using our map of viral-host connections and through characterisation of viral protein sequences, we ranked protein annotations stratified by their predicted host genera. Overall, the proteins were highly enriched for annotations related to viral structural proteins such as baseplate, portal, capsid, head, tail/tail-fibre and tail tape measure but also viral integrase enzymes and Lambda-repressor proteins (Supplementary Data 2). For instance, Lambda-repressor proteins were found in up to ~60% of all viruses suggesting that our dataset was enriched with temperate phages (Fig. 6a). Interestingly, we also identified virally encoded protein domains, which are known to function as viral entry receptors[44], to be enriched within a group of viral populations infecting *Bacteroides* and *Alistipes* such as the TonB plug and TonB-dependent receptor domains (PF07715 and PF00593, Fisher's exact test, adj. $P < 0.05$, FDR-corrected) (Supplementary Data 3). Furthermore, the TonB domains also encode an established immunodominant epitope[45] suggesting that viral populations carry immunogenic entry receptors when expressed

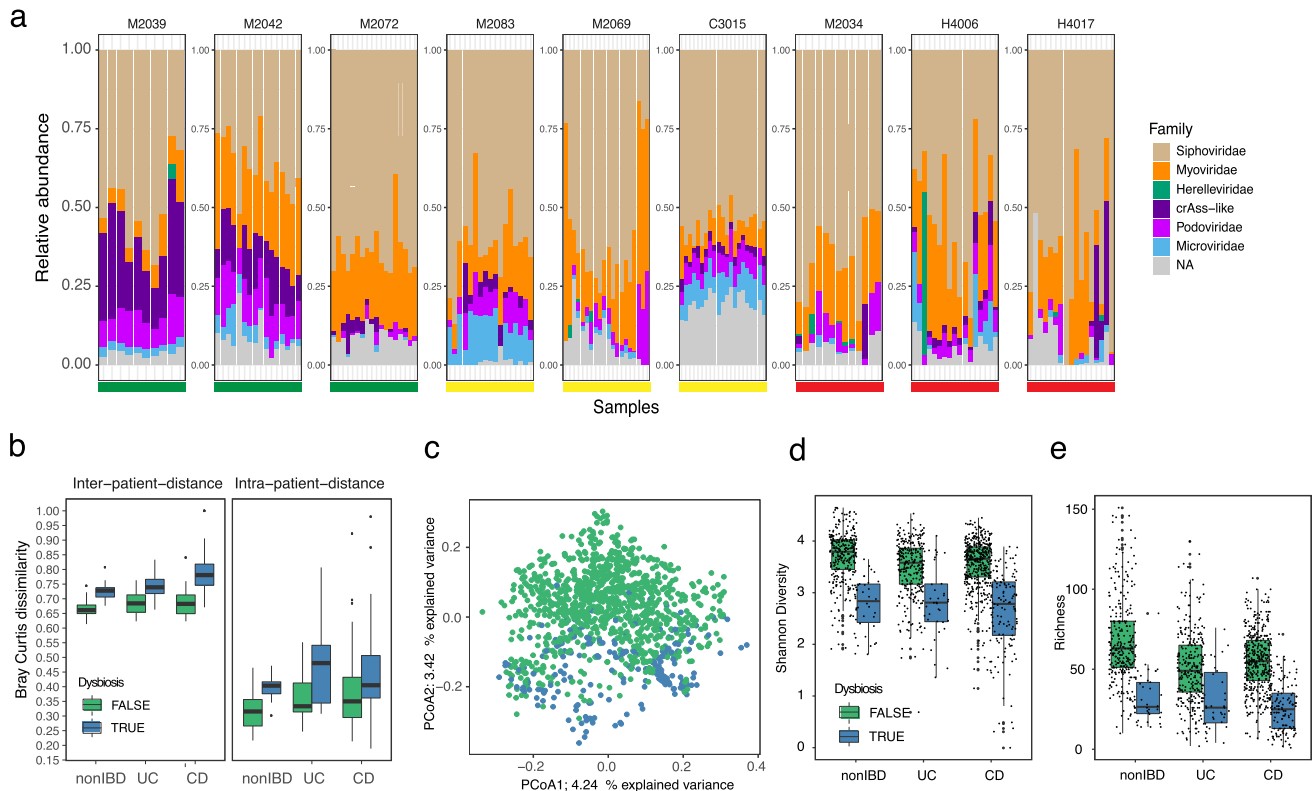

**Fig. 4 The metagenomics estimated virome is personal and highly stable in healthy controls. a** Longitudinal virome compositions for three nonIBD (green bar), three UC (yellow bar) and three CD (red bar) diagnosed subjects. Each panel represents a subject where the virome composition was organised according to the total relative abundance according to the taxonomic viral family, where 'NA' populations are coloured grey. **b** Dissimilarity boxplots based on Bray–Curtis distance (BC) function between samples from different subjects (first panel inter-patient-distance) and between samples from the same subject (second panel intra-patient-distance). The BC distances are shown for samples from nonIBD ($n = 326$), UC ($n = 323$) and CD ($n = 573$) diagnosed subjects. Furthermore, BC distances are coloured according to dysbiosis (blue, UC = 39 samples, CD = 133 samples, nonIBD = 38 samples) or not (green, UC = 284 samples, CD = 425 samples, nonIBD = 286 samples). **c** Principal component analysis (PCoA) of Bray–Curtis distance matrix calculated from the viral abundance matrix in HMP2. Each point is coloured according to diagnosed dysbiosis as in (**b**). **d** Shannon-diversity estimates of metagenomics derived viral populations and coloured according to dysbiosis as in (**b**). **e** Per sample viral population richness based on the number of viral populations detected (abundance >0) in the samples. Coloured according to dysbiosis as in (**b**). nonIBD: healthy control, UC ulcerative colitis, CD Crohn's disease.

by their host. Finally, Reverse Transcriptase (RT, PF00078) proteins were also highly detected, in agreement with recent results[20] and shared by all viral populations irrespective of the predicted host (Supplementary Fig. 18A). These proteins are known modules in bacteriophage diversity generating regions that cause hypervariability in specific viral genes[46].

**Exploring the dark-matter metavirome.** Finally, we investigated the part of the RF predicted bins that did not resemble any of the known genomes, i.e. metagenomics 'dark-matter'. These were defined as populations without at least one HQ or MQ viral bin. Such populations, therefore, represent a part of the microbiome that are not classified as bacterial, archaeal and not alike known viral genomes. Since dark-matter populations were numerous (97.6% of all RF predicted VAMB clusters) we suspected many of these to be fragmented viruses or unknown viruses. Dark-matter populations larger than 10 kbp with at least one viral hallmark gene displayed lower viral prediction scores compared to HQ-MQ viral bins, while bins targeted by CRISPR spacers displayed a significantly higher prediction score (T-test, two-sided, CI = 0.05:0.067, P = 2.2e-16), thus we annotated these as 'viral-like' (Fig. 6b and Supplementary Fig. 19). When stratifying read

abundance on these groups (HQ-MQ, viral-like, dark-matter) we found them to explain on average 2.77, 2.04 and 17.7% of total read abundance across samples, respectively (Fig. 6c). Furthermore, we found that 5% HQ and 3.7% viral-like populations were detected in at least 40% of the patients across disease states. For instance, HQ viral populations cluster 653 were observed in 41% of the cohort (Fig. 6d). Simultaneously, a viral-like population of 1338 was observed in 98% of individuals but displayed a low similarity to any reference genome (Fig. 6e). However, caution should be taken with labelling dark-matter bins as viruses since these are possibly incomplete, contaminated or contain other types of mobile genetic elements that encode proteins shared with viruses such as integrases, polymerases and toxin-antitoxin addiction modules[47,48].

## Discussion
Because of the current challenges facing the viral assembly process, which results in partial and fragmented viral genome recovery[13,15], viral communities have traditionally been notoriously difficult to study. Metavirome datasets have been crucial for identifying a broad scope of viruses, in particular virulent ones. However, the paucity and difficulties in creating metavirome datasets combined with the fact that bulk metagenomes are

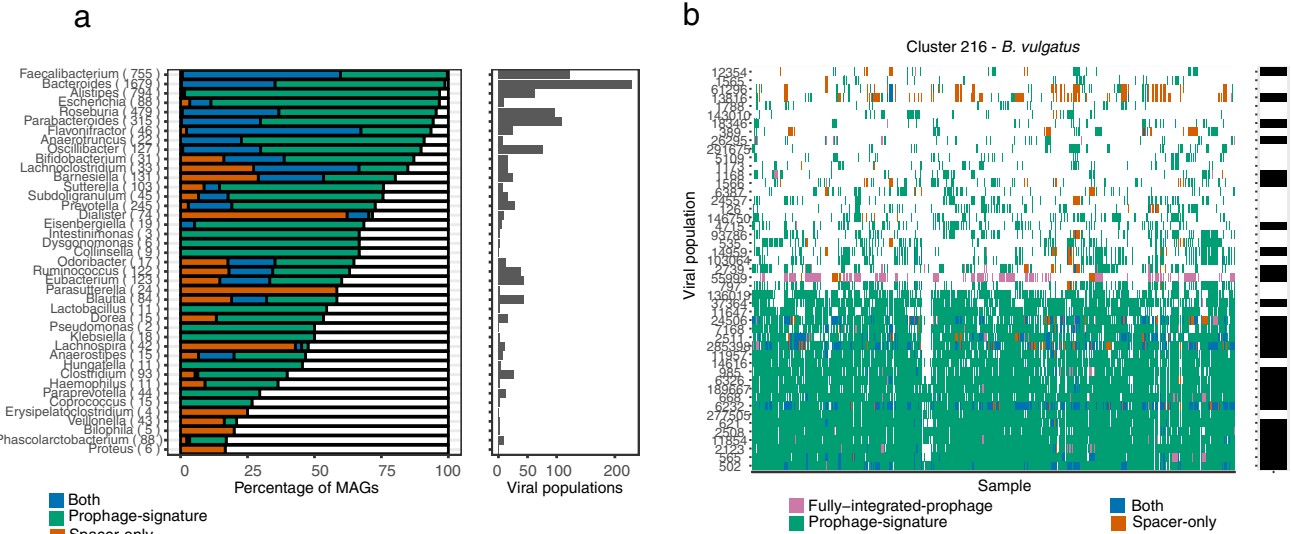

**Fig. 5 Viral–host interactions can be explored from viral populations and MAGs. a** Bacterial MAGs and viral relations. Each MAG was connected to the viral bins using either sequence alignment of the virus to MAG (green), CRISPR spacer alignment (orange) or both (blue). The right panel shows the percentage of MAGs, grouped by genera, that was annotated with the virus via alignment or CRISPR spacer. The number of distinct viral populations associated with a MAG genus based on either of the following: sequence alignment of the virus to a MAG within the given genera, CRISPR spacer alignment or both. **b** Viral association to all MAGs of VAMB cluster 216 (*B. vulgatus*) in the HMP2 dataset. For instance, viral population 502 was associated with the *B. vulgatus* across the vast majority of samples where *B. vulgatus* was present.

produced in abundance, calls for more methods to efficiently extract the viromes found therein. Here we present an improved framework for exploring metavirome directly from bulk metagenomics datasets.

Using our map of viral and bacterial connections we wanted to associate and study the human gut virome along highly abundant gut bacteria such as *Bacteroides* and *Faecalibacterium*. Several of these genera represent not only highly abundant gut commensals but also hotspots for viruses as we have shown by connecting 230 and 123 viral populations to *Bacteroides* and *Faecalibacterium*, respectively. Viral hotspots could be partially explained by factors such as their absolute numbers and genome sequencing depth, which may allow for a more complete assembly of CRISPR-cas systems. A large part of these connections was also made via prophage signatures, i.e. shared genomic elements between bacteria and phage (Fig. 5). Prophage signatures could be the result of increased rates of lysogeny and coinfection as higher microbial densities and phage adsorption rates provide favourable conditions for multiple phages to 'piggyback' highly productive hosts and exchange genetic material[49]. In agreement with other results[11], we found that *F. prausnitzii* genomes are rich in prophages and were able to annotate one for 99.7% of the bacterial bins in HMP2. In the HMP2 cohort, we identified 250 likely temperate *Caudovirales* viruses expanding in a synchronised manner with bacterial hosts following increasing gut dysbiosis[6,10]. However, more work is needed to outline the intricate virus-host dynamics that can explain the degree of viral influence on bacterial perturbations observed in IBD related to dysbiosis such as 'Piggyback-the-Winner' or 'Kill-the-Winner' dynamics[50] with carefully calculated correlations[51].

Based on the viral proteomes it is clear that a majority of HQ viruses extracted in the bulk metagenomes are likely temperate as we have found integrase proteins in 46% of the viral populations and Lambda-repressor proteins in 60% of viruses infecting *Faecalibacterium* bacteria. This adds to the expectation that the non-enriched viromes can be biased toward viruses that infect the dominant host cells in the sample[18]. Interestingly, we found examples of viruses encoding proteins with immunodominant

epitopes such as the TonB plug domain (PF07715) and TonB-dependent beta-barrel (PF00593)[45] in hundreds of viral proteomes extracted from viruses infecting members of *Bacteroidetes* such as *Bacteroides* and *Alistipes*. A recent study has shown that common structural phage proteins such as the tail length tape measure protein (TMP) also harbour immunodominant epitopes that cross-react to cause antitumour immunity[52]. It is therefore interesting to investigate the extent to which viral organisms can influence the human host-microbiota immune balance through horizontal transfer and expression of immunogenic proteins.

Metavirome studies have until now been the primary source for exploring viral diversity in microbiomes. Now, viral populations are increasingly uncovered in bulk metagenomes and we showed that more complete viral genomes can be identified via viral binning across three different cohorts, similar results were found in a recent paper focused on binning of sequenced viral particles[53]. Our approach allowed precise clustering of both viral and bacterial populations in three cohorts that enabled direct investigation into viral-host interactions and discovery of new diversity. We believe that future studies can greatly leverage this approach to conduct virome analyses and investigate the viral influence of the intricate microbiome ecosystem that governs human health.

## Methods

**Datasets**. The Copenhagen Prospective Studies on Asthma in Childhood 2010 (COPSAC) dataset consisted of 662 paired samples obtained at age 1 year from an unselected childhood cohort (refs. [33,34]). The COPSAC study was conducted in accordance with the guiding principles of the Declaration of Helsinki and was approved by the Capital Region of Denmark Local Ethics Committee (H-B-2008-093), and the Danish Data Protection Agency (2015-41-3696). Both parents gave oral and written informed consent before enrolment. The Diabimmune dataset contained 112 paired samples from controls and type 1 diabetes patients. The Human Microbiome Project 2 cohort consisting of 1317 metagenomic samples were downloaded from https://ibdmdb.org/tunnel/public/summary.html.

**Processing of metagenomics and metaviromics datasets**. Metagenomic samples of infants en route T1D recruited to the Diabimmune study were downloaded from https://pubs.broadinstitute.org/diabimmune (October 2019). Metagenomic samples were quality-controlled and trimmed for adaptors using kneaddata

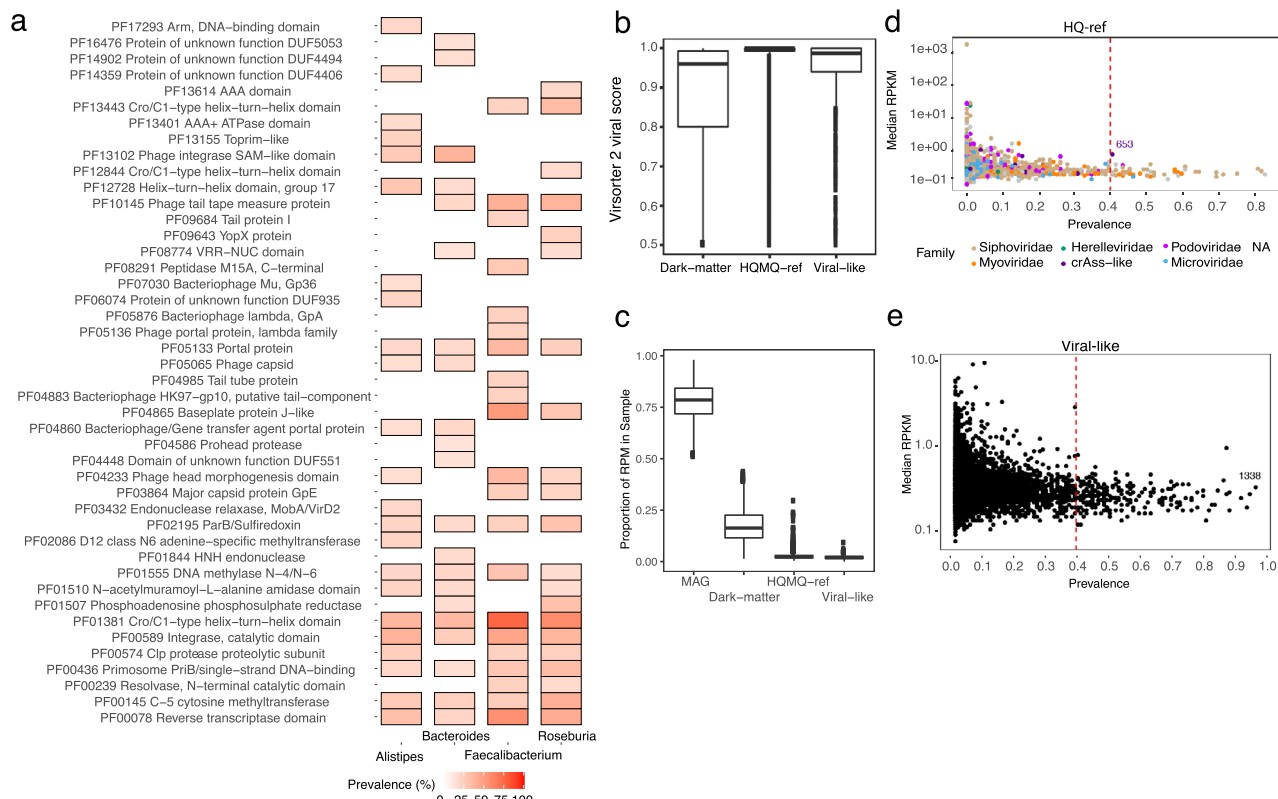

**Fig. 6 Viral proteins and the dark-matter metavirome. a** The percentage of HQ viruses, associated with four bacterial host genera; Alistipes, Bacteroides, Faecalibacterium and Roseburia, which encode top-20 prevalent PFAM domains. **b** Virsorter2 viral prediction scores for all viral bins with at least one viral hallmark gene. Completeness was estimated using CheckV and the bins were grouped as (1) HQ-MQ-ref when completeness ≥50% or high-quality ≥90% ($n = 45,983$ bins), (2) bins with less than 50% completeness were annotated as Dark-matter ($n = 392,226$ bins), and (3) dark-matter bins with confident CRISPR spacers against a bacterial host were annotated as Viral-like ($n = 43,695$ bins). **c** The distribution of sample RPM of bacterial MAGs, HQ-MQ-ref viral populations, Dark-matter and Viral-like populations as defined in (**b**). The majority of sample reads were mapped to MAGs but on average 17.7% of all reads mapped to Dark-matter bins. **d** The abundance in RPKM of rare and highly prevalent viruses with an HQ genome in HMP2. Each point represents a viral population coloured according to the viral taxonomic family. The progenitor-crAssphage is indicated as cluster 653. **e** As in (**d**) but with viral-like populations like cluster 1338 showing that many are low abundant, but highly prevalent. RPM read per million, RPKM read per kilobase million.

(https://github.com/biobakery/kneaddata) and trimmomatic (v.0.36)[54] settings: ILLUMINACLIP: NexteraPE-PE.fa:2:30:10 LEADING:20 TRAILING:20 SLI-DINGWINDOW:4:20 MINLEN:100. Each metagenomic sample was assembled individually using metaspades (v. 3.9.0)[55] using the parameters '--meta, -k 21,33,55,77,99' and filtered for contigs with minimum length of 2000 base pairs. Mapping of reads to contigs was done using minimap2 (v.2.6)[56] using '-N 50' and filtered with samtools (v.1.9)[57] using '-F 3584'. Contig abundances were calculated using jgi_summarize_bam_contig_depths (v.2.10.2)[58]. Metagenomic bins were defined using VAMB (v. 3.0.1)[29] to cluster the metagenomic contigs into putative MAGs and viruses. Initially, the contents of all bins were searched for viral proteins with hmmsearch (v. 3.2.1)[59] against VOGdb (v. 95) (https://vogdb.csb.univie.ac.at/). The presence of bacterial hallmark genes were determined using both CheckM (v.1.1.2)[60] and hmmsearch against the miComplete bacterial marker HMM database (v.1.1.1)[61]. A viral score of each contig was computed using DeepVirFinder (DVF v.1.0)[25]. We initially assessed the metaviromes of the COPSAC and Diabimmune datasets using ViromeQC[62] and found 5.1 and 0.21 times viral enrichment of the two datasets, respectively (Supplementary Fig. 1).

**Training the random forest to predict viral bins**. First we established an initial viral truth set in the metagenomic assembly for the random forest classification. For each metagenomics bin, we computed the fraction of contigs mapping to a set of non-redundant viral sequences (Gold standard) using blastn (v. 2.8.1)[63] with a minimum sequence identity of 95% and query coverage of 50%. Gold standard viral sequences of the paired metaviromics datasets were provided by the authors of the Diabimmune and COPSAC studies (https://doi.org/10.5281/zenodo.5821973). Metagenomic bins with ≥95% of contigs matching with the above criteria were annotated as Viral bins. For annotating bacterial bins, MAGs were identified using CheckM (v.1.1.2). MAGs with a completeness score of 10% or above and contamination ≤30% were added to the training and validation set labelled as bacteria. For training, we used COPSAC and validated using the Diabimmune dataset. Thus the model was trained to distinguish confidently labelled bacterial and viral bins

produced by VAMB, this provided an RF model highly effective at removing non-viral bins and providing a highly enriched candidate set of viral bins that could be further evaluated using dedicated validation tools. In the RF model we included features such as bin size, the number of distinct bacterial hallmark genes, the number of different PVOGs in a bin divided by the number of contigs in the bin, viral prediction DVF score (median DVF score for a bin) defined by DeepVir-Finder. The Random Forest model was implemented in Python using *RandomForestClassifier* (sklearn v. 0.20.1) with 300 estimators and using the square root of the number of features as the number of max features. The model was trained on the COPSAC dataset using 40% of observations for training and 60% for validation. Subsequently ROC/AUC, recall and precision was calculated using the Diabimmune recovered viruses as an evaluation set. We ran viral predictions on contigs of minimum 2,000 bp using Virsorter2 (v. 2.2.3)[30], viralVerify (v.1.1)[31], Seeker (v.1.0)[64], Virfinder (v.1.1)[26] and DeepVirfinder (v. 1.0), all on their default settings. In order to calculate single-contig viral prediction performance, a contig was labelled viral if the prediction score was above 7, 0.5, 0.9, 0.9 and 0.9 viralVerify, Seeker, Virfinder, DeepVirFinder and Virsorter2, respectively. Genome level predictions (bacterial or viral) for each of the aforementioned tools were done with the same cutoffs mentioned above but based on the aggregated bin-score. The bin-scores were aggregated as a contig-length weighted mean, mean and median.

**Virus binning and prediction performance on simulated datasets**. We compared the viral binning performance of VAMB and MetaBAT2 using the official CAMISIM method to create assemblies and metagenome profiles[65]. To this end we generated three different metagenome compositions with up to 308 reference genomes; one mixed with bacteria, plasmids and viruses to test binning in complex samples i.e. high diversity (1), one with only crass-like viruses to test binning with highly similar viruses i.e. high relatedness (2) and a set of small viruses (<6000 bp) including members of the Microviridae family to address the bias of size (3). Bacterial genomes were pulled from NCBIs refseq genome repository 2021, plasmids from the PLSDB database (v. 2021_06_23)[66] and viral genomes from the recent MGV database[20] (Supplementary Data 4). Fragmented genome assemblies

were generated for each metagenome composition using CAMISIMs (v.1.1.0) metagenome simulation-pipeline with default settings for ten samples[65]. In order to test genome recovery via binning, abundance of the simulated contigs were calculated by mapping of reads to contigs with minimap2 (v.2.6) using '-N 50' and filtered with samtools (v.1.9) using '-F 3584'. Then the abundances were calculated using jgi_summarize_bam_contig_depths from MetaBAT2 and used as input for VAMB and MetaBAT2 that were run with default parameters on the simulated contigs of a minimum of 2000. Furthermore, we ran viral predictions on contigs of minimum 2000 bp using Virsorter2 (v. 2.2.3)[30], viralVerify (v1.1)[31], Seeker (v.1.0)[64], Virfinder (v.1.1)[26] and DeepVirfinder (v. 1.0), all on their default settings. In order to calculate single-contig viral prediction performance, a contig was labelled viral if the prediction score was above 7, 0.5, 0.9, 0.9 and 0.9 viralVerify, Seeker, Virfinder, DeepVirFinder and Virsorter2, respectively. Genome level predictions (viral or non-viral) for each of the tools were done with the same cutoffs mentioned above on the aggregated bin-score. The bin-scores were aggregated as a contig-length weighted mean, mean and median. The RF model was run as intended where information about each contig was aggregated and parsed by the model to produce a viral/non-viral label. Optimised and overfitted bin/genome-score thresholds were determined by inspection of genome-score distributions (Supplementary Fig. 5) for each viral prediction method. These thresholds were −1.3, 0.75, 0.9, 0.5 and 0.5 for viralVerify, Seeker, Virsorter2, DeepVirFinder and Virfinder, respectively.

**Intersection of viruses in MGX and MVX data**. In order to identify the number of viruses assembled and binned in the metagenomic (MGX) datasets we searched the metavirome (MVX) viruses in all-vs-all search and calculated genome-to-genome average nucleotide identity (ANI) and genome coverage as an aligned fraction (AF). Here we defined species level above 95% ANI and strain-level above 97% ANI. Overlapping or also described as highly-similar viruses between the paired MGX and MVX datasets were those fulfilling the ANI >95% and >75% AF criteria. This search was conducted using FastANI (v.1.1, '-fragmenlen 500 -minimumfrag 2 -minimum 80% ANI')[67] with genome coverage ≥50% (bidirectional fragments / total fragments). We note that hits with less than 80% ANI were not included. We expected that we might be able to find fragmented/incomplete viruses assembled in the metavirome but were more curious about near-complete viruses, thus we quality controlled all MVX viruses using CheckV (v0.4.0, default settings, database v. 0.6)[28] to achieve a completeness estimate for each. By labelling the quality of each MVX virus we organised the success of genome recovery into the four CheckV levels (low-quality ≤50%, medium-quality ≥50%, high-quality ≥90%, Complete = closed genomes based on direct terminal repeats (DTR) or inverted terminal repeats (ITR)). Furthermore, we also quality controlled the putative viruses assembled and binned in the MGX to ask the reverse question, i.e. to what extent do we find complete viruses with no similarity to viruses in the MVX.

**Completeness of viruses recovered in metavirome and bulk metagenomes**. To standardise our viral recovery performance across different datasets, we used the guidelines on Minimum Information about an Uncultivated Virus Genome (MIUViG)[18]. The viral completeness of viruses from metaviromics data was assigned using CheckV described as above. CheckV was used to conduct a benchmark on virus genome completeness by evaluating single-contig assemblies against the use of viral bins (also described as viral MAGs). To this end, we based our analysis solely on AAI-model predictions. As the CheckV authors note, the method was not designed by default to accommodate viral MAGs and may not deal properly with contaminants from bacterial or viral sources[28]. This became clear as we observed a majority of HMM-model predicted viruses consisting of sequences with close to zero percent viral sequence (Supplementary Fig. 20). We suspect that this was to be expected since the HMM-model was designed for single-contig viral assemblies. Thus, the model could not deal properly with cases where a viral marker gene was identified in a single-contig of the bin and contaminating sequences inflate the total bin size to randomly fit into the reference size range of viruses encoding the same viral marker. Hence to avoid including false-positive viral bins, we defined a bin as HQ-ref when at least one bin in the VAMB cluster contained an HQ evaluation based on AAI-evaluation. All viral bins with a CheckV computed genome copy number ≥1.25 were removed to control for 'concatemers'. Finally, viral bins with an estimated completeness >120% (over-complete-genomes) were removed as well to control for highly contaminated bins. We found that the frequency of HQ genomes, which according to MIUViG standards[18,19] were 'overcomplete-genomes' (estimated completeness >120%), was between 7.9–14.2% for the viral bins and 3.8–6.1% for single-contig evaluation (Supplementary Table 2). Hence, the binning approach generates more over-complete-genomes, although these can be identified and removed using for instance CheckV, which we highly advise. We found that after removal of over-complete-genomes, VAMB mainly produces viral bins with low contamination and high purity. Contamination and purity in this case was calculated according to a reference/ground truth. Example: for a viral bin with a total size of 90,000 and 8000 bp not aligned to the corresponding ground truth genome, contamination is 8000/90,000 = 8.8% and purity is 100–8.8% = 91.2%. The remaining populations without a single HQ or MQ bin within their VAMB cluster were described as dark-matter. For identifying viruses in 'dark-matter' populations, we ran Virsorter2

(v.2.0)[30] and considered sequences or bins with a prediction score >0.75, at least one viral hallmark and a minimum size of 10 kbp as a putative virus. In this subset of putative viruses, we defined 'viral-like' dark-matter when they were targeted with a CRISPR spacer by a bacterial MAG (see 'Viral-host prediction').

**Viral taxonomy and function**. While the databases of viral genomes continue to grow, taxonomy is still a challenge for viral genomes with little similarity to the International Committee on Taxonomy of Viruses (ICTV) annotated genomes. Viral proteins were predicted using prodigal (v.2.6.3)[68] using '-meta'. All proteins were annotated using viral protein-specific databases such as VOG (http://vogdb.org) or viral subsets of TrEMBL used in the tool Demovir (v.1.1.0) (https://github.com/feargalr/Demovir). Viral taxonomy was assigned to each bin using the plurality rule described before in Roux et al. (ref. [19]): (1) taxonomy was assigned to genomes with at least two PVOG proteins using a majority vote (≥50% else NA) on each taxonomic rank based on the last common ancestor (LCA) annotation from the PVOG entries. (2) The CheckV VOGclade taxonomy was transferred if available from the best viral genome match in the CheckV database. In order to annotate 'crAss-like' viruses, predicted proteins were aligned using blastp (v. 2.8.1)[63] to the large subunit terminase (TerL) protein and DNA polymerase (accessions: YP_009052554.1 and YP_009052497.1) of the progenitor-crassphage using already described cutoffs[69]. When investigating taxonomic annotations, considering only MQ-Complete viral bins, the most dominant viral family annotated was Siphoviridae accounting for 53.5% of the viral bins (Supplementary Figure 9). Furthermore, we could assign Myoviridae 14.57%, Podoviridae 8.59%, Microviridae 8.30%, crAss-like 3.61%, CRESS 2.52%, Herelleviridae 1.37% and Inoviridae 0.58%. Finally, 6.93% of viruses could not be confidently assigned any viral taxonomy. Similar distributions of taxonomic annotations were also observed for Diabmmune and COPSAC (Supplementary Table 3).

For viral proteomes, we utilised CheckV's contamination detection workflow to extract proteins encoded only in viral regions to avoid host contamination. These viral proteins were analysed with interproscan (v. 5.36-75.0)[70] using the following databases: PFAM, TIGRFAM, GENE3D, SUPERFAMILY and GO-annotation. For each annotated functional domain in viruses predicted to infect a given host genus enriched proteins were identified using Fisher's exact test using the function *phyper* in base R. P-values were adjusted using false discovery rate (FDR) correction[71]. Viral reverse transcriptase enzymes were grouped into DGR-clades by querying each protein sequence against a database of RT DGR clade HMM models while DGR target genes were identified using the methods and pipeline provided[72].

**Phylogenetic tree of crAss-like viruses**. A phylogenetic tree was constructed for crAss-like viruses identified in the HMP2 dataset based on proteins annotated as the large terminase subunit protein (the TerL gene). First, viral bins annotated as 'crAss-like' proteomes were determined as described above. 'crAss-like' proteomes were aligned to a terminase large subunit protein (accession: YP_009052554.1) and also against VOGdb hmmsearch (v. 3.2.1, hmmscore ≥ 30)[59] against VOGdb (v. 95) (https://vogdb.csb.univie.ac.at/). The VOG entries corresponding to the terminase large subunit:VOG00419, VOG00699, VOG00709, VOG00731, VOG00732, VOG01032, VOG01094, VOG01180 and VOG01426, were identified using a bash command on a VOGdb file: 'grep -i terminase vog.annotations.tsv'. An alignment file was produced for proteins, annotated as terminase large subunit, using MAFFT (v. 7.453)[73] and Trimal (v. 1.4.1)[74] and converted into a phylogenetic tree using IQtree (v. 1.6.8 -m VT + F + G4 -nt 14 -bb 1000 -bnni)[75].

**Viral-host prediction**. Viral genomes were connected to hosts using a combination of CRISPR spacers and sequence similarity between viruses categorised as HQ-ref and MAGs. CRISPR arrays were mined from COPSAC and HMP2 MAGs using CrisprCasTyper (v.1.2.3)[76] with '--prodigal meta' and all spacers were blasted with blastn-short (v. 2.8.1)[63] against all viral genomes to identify protospacers. CRISPR spacer matches with ≥95% sequence identity over 95% of spacer length and maximum of two mismatches were kept. In order to identify the host of viruses, viral bins were aligned to MAGs using FastANI (v.1.1, '--fragLen 5000 --minFrag 1')[67] and blastn megablast (v. 2.8.1)[63] with a minimum ANI ≥90% and sequence identity ≥90, respectively. We followed the approach described by Nayfach et al. (ref. [42]) to calculate host-prediction consensus and accuracy. The viral host was defined using a plurality rule at each taxonomic rank based on the lineage of bacteria connected using either CRISPR spacer or alignment to the given virus. The cutoffs described above were selected after benchmarking the alignment approach with FastANI and blastn at various thresholds. We observed an increased host-prediction consensus and accuracy at the species rank using the threshold described above with FastANI with ANI ≥90% based on at least one 5000 bp fragment, compared to blastn thresholds described by Nayfach et al. (ref. [42]). We evaluated the agreement of our two host prediction methods and found up to 58% consensus on host taxonomy on species rank (Supplementary Fig. 11A). We further benchmarked host-prediction purity by calculating the most common host for each viral population according to (1) CRISPR spacer and (2) alignment independently.

Viruses were annotated as temperate virus if (1) the virus was found to be integrated into a MAG with ≥80% query coverage and ANI ≥90% or (2) an integrase protein-annotation could be found in the viral proteome. Integrase

proteins were determined by searching for *integrase* in the InterPro entry description of each interproscan protein-annotation (see Viral taxonomy and function for details).

**Differential abundance of viral populations and MAGs.** Sample abundance of each viral population was calculated as a mean read per kilobase million (RPKM) of all contigs with at least 75% coverage belonging to a VAMB cluster. Differential abundance analysis of all viruses was tested using the Linear-mixed-effect model R-function *lmer* (lme4 package v. 1.1-26)[77]. The model used was 'Virus ~ dysbiosis_index + diagnosis + sex + (1|Subject)'. Subjects were included as random effects to account for the correlations in the repeated measures (denoted as (1 | subject)) and the log-transformed relative abundance of each virus was modelled as a function of diagnosis (a categorical variable with nonIBD as the reference group) and the dysbiosis index (continuous covariate) while adjusting for subjects age as a continuous covariate and sex as a binary variable.

**Definition of boxplots.** The lower and upper hinges correspond to the first and third quartiles (25th and 75th percentiles). Centre corresponds to the median. The upper and lower whiskers extend from the hinge to the highest and lowest values, respectively, but no further than 1.5 × interquartile range (IQR) from the hinge. IQR is the distance between the first and third quartiles. Data beyond the ends of whiskers are outliers and are plotted individually. This definition is used for all main and supplementary figures displaying a boxplot.

## Data availability

The Diabimmune dataset and HMP2 datasets are available from the European Nucleotide Archive with the accessions PRJNA387903 and PRJNA398089. The COPSAC metagenomics and metaviromics datasets are available with the accessions PRJNA715601 and PRJEB46943, respectively. Gold standard virus genomes for COPSAC and Diabimmune were provided by Shiraz Shah and Tommi Vatanen, respectively, and are available on Zenodo: https://doi.org/10.5281/zenodo.5821973. A CodeOcean capsule of PHAMB v.1.0, including a dataset of 3,000 contigs from 5 HMP2 samples, is available at CodeOcean (https://doi.org/10.24433/CO.4597219.v1). Furthermore, the capsule includes a Dockerfile encoding required databases, Python modules, Snakemake and DeepVirFinder dependencies. Genomes used in the viral CAMISIM benchmark have been uploaded to Zenodo and are available here: https://doi.org/10.5281/zenodo.5821973. Simulated genomes are listed in Supplementary Data 4, entries were collected from the PLSDB database (v. 2021_06_23), MGV database (2021), NCBI Refseq (May 2021). Source data is provided with this paper. Source data are provided with this paper.

## Code availability

The VAMB code is available at https://github.com/RasmussenLab/vamb and the PHAMB workflow is available at https://github.com/RasmussenLab/phamb.

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

## Acknowledgements

We thank Mani Arumugam, Eduardo Rocha, Nicolas Rascovan, Ramnik Xavier and Hera Vlamakis for fruitful discussions. J.J., J.N.N. and S.R. were supported by the Novo Nordisk Foundation (grant NNF14CC0001). COPSAC authors were supported by The Lundbeck Foundation (Grant no R16-A1694); The Ministry of Health (Grant no 903516); Danish Council for Strategic Research (Grant no 0603-00280B) and The Capital Region Research Foundation have provided core support to the COPSAC research center.

## Author contributions

S.R. conceived the study and guided the analysis. J.J. wrote the software, performed the analyses and wrote the manuscript. S.A.S., J.S., L.D., and D.S.N. generated metavirome data and created the viral gold standard for COPSAC data. S.J.S. and J.S. generated COPSAC metagenome data. D.R.P. and J.N.N. guided the analyses. J.J., S.R., M.L.J. and D.R.P. wrote the manuscript with contributions from all co-authors. All authors read and approved the manuscript.

## Competing interests

The authors declare no competing interests.
