## [Peer Review File · Nature Communications]

REVIEWER COMMENTS

Reviewer #1 (Remarks to the Author):

Johansen et al. present VAMB, the first and only tool dedicated to metagenomic binning of viruses. I found the manuscript to be very timely and believe that it will be of broad interest in the field. I have several comments and concerns however (as outlined below) regarding the benchmarking of VAMB and comparison to existing approaches.

Major comments:

VAMB is a two-step approach consisting of binning with PHAMB and classification of the subsequent bins (virus vs microbial). For the former step, it would be interesting to see how VAMB compares with other commonly used binning tools. Towards this goal, the authors could apply VAMB and other tools to a mock dataset of entirely viral genome fragments and compare the binning tools in terms of sensitivity and precision. The authors could also modify properties of the simulation, including viral diversity and viral relatedness to determine their effect on performance.

For benchmarking of bin classification (virus vs microbial), it's important to use simulated data and to evaluate several viral prediction tools. Currently only DVF is evaluated. Other viral prediction tools could include Virsorter2, viralVerify, VIBRANT, and Seeker. Likewise, the authors should explore the best way to combine viral scores across contigs in each bin to give these methods the best chance at optimal performance (e.g. mean score, median score, with or without length-weighting).

Related to the last point, how does VAMB handle plasmid sequences? Distinguishing between viruses and plasmids can be challenging, particularly if plasmids are not included in the training dataset.

It's a great idea to compare viral genomes from single-contig assembly versus binning. However, I found some issues with the analysis in Figure 1C, in which the authors compare the number of viruses recovered with DVF (single-contig) versus VAMB (viral-bins). It's hard to understand whether the difference is because the VAMB viral classifier is better than DVF (shown in 1B), VAMB bins are more complete (due to containing multiple contigs), or VAMB bins are larger due to contamination (since the ground truth is not known). For example, a more accurate viral prediction tool (Virsorter2 or viralVerify) might perform considerably better than DVF and close the gap with VAMB. These issues could be addressed by using simulations instead of real data (as suggested above) and then

compare the number of viral MAGs to the number of viral contigs at a minimum % completeness and % contamination level. This would directly compare the results of binning with VAMB to single contigs where the ground truth is known.

Do the authors have a way to estimate or flag bins that contain contamination from other viruses or from bacteria? While 83% of viral bins matching the gold standard dataset were free of contamination, 17% of these WERE contaminated. An approach like viralVerify could be used to remove contigs labeled as bacteria if proteins were searched versus the Pfam database. Or contigs could be removed that contain many CheckV host markers and no CheckV viral markers or hits to VOGDB. Estimating viral contamination seems more challenging but would also be extremely helpful. For example, if a bin contained a 45 Kb circular viral contig, then any other contig in the bin is likely contamination. Contamination in the bins will lead to overestimation of completeness and problems with data interpretation.

Minor comments:

Does VAMB have biases against certain types of viruses? For example, some viral genomes are quite small (e.g. <5Kb) like Microviruses, Inoviruses, CRESS-DNA viruses, RNA viruses, and others.

Lines 155-157: "Based on 1,340 viral bins that were highly similar to metavirome viruses in COPSAC (see methods), we found in 83.7% of all cases, every contig in the bin mapped to the virus". I assume all these bins contained at least 2 contigs? Can you confirm?

Line 243: "none of the 916 crAss-like bins could be associated with any of the 3,306 Bacteroidetes bins in our dataset, which suggests that crAss-like phages are not frequently targeted by CRISPR spacers extracted from Bacteroidetes CRISPR-Cas systems". I thought this was interesting, but I wonder how generalizable this result is. Another paper found the opposite result: <https://doi.org/10.1038/s41467-021-21350-w>. Were you able to assemble a large number of CRISPR arrays from the Bacteroidetes bins? Do these crAss-like bins match CRISPR spacers from reference genomes?

Did binning allow you to recover genomes of any very large viruses? What was the size of the largest viral bins, and are they likely to represent pure viral genomes?

Line 263: what is the significant of the reverse transcriptase genes?

The authors identify "hotspots for viruses". Might these patterns be partially explained by (i) the prevalence of the CRISPR-cas system, and the (ii) total number of MAGs from these groups?

Line 284: The authors conclude that "evidently, a significant portion of the sequenced microbiome remains dark-matter while HQ viruses identified in this study only accounts for a small fraction of the sequenced space.". I feel this conclusion was not adequately supported from the data. Dark matter bins (not matching and reference virus) could result from a number of technical artifacts, including (i) short incomplete bins, (ii) contaminated bins, (iii) non-viral bins. I believe the CheckV database includes a very large number of human gut viruses and has very good coverage of the gut virome, so I suspect these technical issues might be having an outsized impact.

Reviewer #2 (Remarks to the Author):

The authors describe a method and its application for thorough analysis of the virome from metagenomic samples -- both those enriched for viruses and those directly sequencing. This manuscript is well done. It is clearly explained and contains adequate detail. The approach is noteworthy as other methods classify individual contigs to classify but this includes a binning step and training of a RF model to classify the bins.

The authors point to GitHub accounts that contain their code and work flow. My one critique is that the GitHub account or the workflow is incomplete (some steps are noted as "in progress"). Ideally this would be complete, as it would provide assistance to those who wish to use the method that the authors developed.

First of all we want to thank the reviewers for their time and efforts in reviewing our manuscript. We appreciate the comments and suggestions, and have performed several experiments and added new analyses to the manuscript to substantiate the performance of our viral binning methods. We especially focused on binning performance, viral predictive performance and viral bin contamination. Furthermore, we have revisited results on “Jumbo” viruses and crAss-like viral host annotation.

The most notable addition and changes to the manuscript are:

New datasets and experiments:

1. With the CAMI consortium's official pipeline we simulated viral, plasmid and bacterial genome assemblies and populations. Here we found VAMB overall superior to MetaBAT2 in terms of viral-binning performance on the total number of genomes and at higher levels of precision and recall.
2. Based on the same simulated data we benchmarked several virus prediction tools (Virsorter2, Virfinder, Deepvirfinder, Seeker and Viralverify) against each other on ground-truth annotated viral contigs and on genome level against the Random Forest model of the original manuscript submission.

Additional analyses regarding the technical aspects of genome binning:

3. We established low degrees of viral bin contamination in both real data using vOTUs from COPSAC dataset as truth and in the simulated data.
4. We found no drop in binning performance on small virus genomes relative to larger virus genomes using simulated data.
5. We analysed if plasmid bins can be differentiated downstream from viral genomes using CheckV.
6. Analysis on reference genome alignment consensus using evaluation tools such as CheckV.

Other analyses with changes in the revised manuscript:

7. Updating annotation numbers of crAss-like viruses in HMP2 and comparison to a large CRISPR-spacer database.
8. Provided a table of all putative “Jumbo” viral bins across all bulk metagenomic datasets including HMP2, Diabimmune and COPSAC.

An example of running the PHAMB Random forest model can be found in the

CodeOcean capsule: Passed on to the Reviewers by CODEOCEAN

Simulated CAMI data and genomes can be found on a Zenodo repository:

<https://doi.org/10.5281/zenodo.5676246>

Golden standard viral genomes are also provided as a Supplementary data 4 (zip) file

Furthermore, we provided source data for all main figures and simulation results.

Johansen et al. present VAMB, the first and only tool dedicated to metagenomic binning of viruses. I found the manuscript to be very timely and believe that it will be of broad interest in the field. I have several comments and concerns however (as outlined below) regarding the benchmarking of VAMB and comparison to existing approaches.

Major comments:

VAMB is a two-step approach consisting of binning with PHAMB and classification of the subsequent bins (virus vs microbial). For the former step, it would be interesting to see how VAMB compares with other commonly used binning tools. Towards this goal, the authors could apply VAMB and other tools to a mock dataset of entirely viral genome fragments and compare the binning tools in terms of sensitivity and precision. The authors could also modify properties of the simulation, including viral diversity and viral relatedness to determine their effect on performance.

Thank you very much for this suggestion to use simulated data to assess the ability of VAMB to bin phages from metagenomics data and our Random Forest model to predict phage bins. We used a total of 308 reference genomes and CAMISIM from the CAMI consortium (Fritz*, Hofmann*, et al. Microbiome 2019) to simulate three different datasets of metagenomics assemblies and abundance profiles. Dataset A contained a mixture of bacteria (N=8), plasmids (N=20) and viruses (N=280) to test binning in complex samples, i.e. high diversity. Dataset B contained only crAss-like viruses (N=80) to test binning with highly similar viruses i.e. high relatedness. Dataset C contained small-viruses (N=50, <6,000 bp) of the microviridae family to address the bias of size. Bacterial genomes were sampled from the Refseq genome repository 2021, plasmids from the PLSDB database and viral genomes from the recent MGV database (Nayfach, et al. Nature Microbiology 2021). For each dataset we generated 10 samples and prepared a benchmark-setup where Recall (sensitivity) and Precision (specificity) of each genome could be calculated based on the CAMI-simulated reference genomes. See the new Supplementary Data 3 for an overview of genomes and accessions used for the simulations. The datasets are available for download at the Zenodo (<https://doi.org/10.5281/zenodo.5676246>) and we added the following in the main text:

Lines 139-144 in Results section: “We then investigated viral-binning performance of VAMB and the prediction performance with simulated datasets including two pure viral and one mixed dataset containing bacteria, plasmids and viruses. The two pure viral datasets comprised 80 crAss-like viruses and 50 small-genome (<6,000bp) random sampled from the MGV database (Nayfach et al. 2021). To establish the mixed dataset, the crAss-like and small-genome datasets were combined with an additional 150 random virus genomes, 8 bacterial genome isolates and 20 plasmids (see methods).”

Lines 459-487: in Methods section: “We compared the viral binning performance of VAMB and MetaBAT2 using the official CAMISIM method to create assemblies and metagenome profiles (Fritz et al. 2019). To this end we generated 3 different metagenome compositions with up to 308 reference genomes; one mixed with bacteria, plasmids and viruses to test binning in complex samples i.e. high diversity (1), one with only crass-like viruses to test binning with highly similar viruses i.e. high relatedness (2) and a set of small-viruses (<6,000 bp) including members of the Microviridae family to address the bias of size (3). Bacterial genomes were pulled from NCBI's refseq genome repository 2021, plasmids from the PLSDB database (v. 2021_06_23) and viral genomes from the recent MGV database (**Supplementary Data 5**). Fragmented genome assemblies were generated for each metagenome composition using CAMISIMs metagenome simulation-pipeline with default settings for 10 samples. In order to test genome recovery via binning, abundance of the simulated contigs were calculated by mapping of reads to contigs with minimap2 (v.2.6) using ‘-N 50’ and filtered with samtools (v.1.9) using ‘-F 3584’. Then the abundances were calculated using jgi_summarize_bam_contig_depths from MetaBAT2 and used as input for VAMB and MetaBAT2 that were run with default parameters on the simulated contigs of minimum 2,000. Furthermore, we ran viral predictions on contigs of minimum 2,000 bp using Virsorter2 (v. 2.2.3), ViralVerify (v.1.1), Seeker (v.1.0), Virfinder (v.1.1) and DeepVirfinder (v.1.0), all on their default settings. In order to calculate single-contig viral prediction performance, a contig was labelled viral if the score>0.75 (score from 0-1) for every tool except for ViralVerify where a cutoff of >5 was applied. Genome level predictions (viral or non-viral) for each of the aforementioned tools were done with more stringent cutoffs on the aggregated bin-score, >0.9 for all tools except for ViralVerify where the cutoff was >15. The bin-scores were aggregated as a contig-length weighted mean, regular mean and median. The RF model was run as intended where information about each contig was aggregated and parsed by the model to produce a viral/non-viral label.”

As suggested, we investigated the binning performance of VAMB on viruses compared with MetaBAT2. Overall, we found that VAMB outperformed MetaBAT2 on the mixed genome set on bins with high Recall and Precision (>0.9), 144 vs 134 bins. Interestingly, VAMB captured a much higher number of bins with high Precision (>0.9) but at lower Recall (>0.5). This could suggest that in the simulated data VAMB performs better on some organisms relative to MetaBAT2. On bins with very high levels of Recall (>0.99) MetaBAT2 slightly outperformed VAMB however this was at low Precision (>0.5), 143 vs 127 bins. None of the methods binned more than 50% of all genomes with high recall and precision (144 for VAMB, 134 for MetaBAT2). These results are now presented in the new Supplementary Figure 3A-C which are shown below. Furthermore we updated the main text:

Lines 144-150: “On the mixed dataset, VAMB outperformed MetaBAT2 on bins with high >0.9 recall and >0.9 precision with a total of 144 vs 134 bins, corresponding to just above 50% (144/280) of all simulated virus genomes (**Supplementary Figure**

3a). Furthermore, we found that VAMB binned increasingly a higher number of bins at lower recall (>0.5) and increasing precision levels. Regarding plasmids, both tools were comparable and binned up to 10/20 plasmids with >0.5 recall and >0.95 precision (**Supplementary Figure 3b).**”

A

B

C

D

Supplementary Figure 3. Evaluating VAMB and MetaBAT2 for binning phages on simulated data. A) The number of genomes binned by VAMB and MetaBAT2 with increasing Recall and Precision at three levels 0.9, 0.95, 0.99, separated by organisms viral, plasmid and bacteria. B) The number of genomes binned by VAMB and MetaBAT2 with increasing Recall and Precision > 0.95 for viruses, plasmids and bacteria in their own panel. Heatmap showing the number of genomes binned by VAMB (C) and MetaBAT2 (D) in the Recall and Precision range [0.3-0.99].

When we investigated the two dataset containing virus only genomes, crAss-virus (dataset B) and small-virus (dataset C), VAMB achieved almost perfect binning with 48/50 and 70/80 bins with high Recall and Precision (>0.99). Here, we did not find size of the virus to have a notable influence on binning-performance, nor in the mixed genome set where 80.5% of small viruses were binned with F1>0.9. These results are now presented in the new Supplementary Figure 4 and in the results section:

Lines 150-158: “Next, we addressed how binning performance could be influenced by virus genome size and highly similar viruses. For this we sampled smaller virus genomes (<6,000 bp, n=50) and viruses of the same family (crAss-like, n=80). A total of 48/50 and 70/80 genomes were binned with >0.99 recall and >0.99 precision for the small-virus and same family-virus set, respectively (**Supplementary Figure 4ab**). The ease of binning small viruses was confirmed in the mixed dataset where VAMB captured the majority of small viruses with high recall and precision (F1>0.9) (**Supplementary Figure 4c**), indicating that genome-size was less confounding to binning performance.”

Supplementary Figure 4. Performance of VAMB binning the phage only simulated datasets and small-viruses. A) Precision and recall performance of VAMB binning on small phage genomes (<6,000 bp, n=50). B) Precision and recall performance of VAMB binning on crass-like (n=80) viruses. C) F1-score distributions of small virus genomes and larger virus genomes (>6000 bp) in the mixed genome dataset.

For benchmarking of bin classification (virus vs microbial), it's important to use simulated data and to evaluate several viral prediction tools. Currently only DVF is evaluated. Other viral prediction tools could include Virsorter2, viralVerify, VIBRANT, and Seeker.

As suggested, we have now performed a test of multiple viral prediction tools including Virsorter2, ViralVerify, Seeker, Virfinder and DeepVirfinder. We did not choose to include Vibrant nor CheckV in the benchmark as these tools do not provide a general viral-prediction score pr. contig but rather a factor describing

overall genome quality, which is more relevant for downstream completeness and quality evaluation.

The viral prediction tools were tested on single contig-level using the CAMI simulated assemblies from the mixed set of genomes (dataset A: bacteria+plasmid+virus) described above. We applied a slightly conservative viral-score cutoff considering a contig viral if the viral-prediction-score > 0.75 (score from 0-1) except for ViralVerify where a cutoff of 5 was applied. In case a prediction tool did not provide a prediction for a contig we imputed a score of zero i.e. non-viral. Using these cutoffs we found that DeepVirfinder performed best overall with highest AUC and MCC and second best on F1. Interestingly, DeepVirfinder was only marginally better than the original Virfinder. These results are now shown in Supplementary Figure 5 A.

Likewise, the authors should explore the best way to combine viral scores across contigs in each bin to give these methods the best chance at optimal performance (e.g. mean score, median score, with or without length-weighting).

In order to make a fair benchmark on Genome/Bin-level viral prediction, we used the ground truth labels from the CAMI simulation to test the performance of each viral predictor. Following the advice, we explored different ways of combining viral scores across contigs in each bin for the viral prediction tools (median, mean and contig-length weighted score). For this analysis we applied much more stringent thresholds on the aggregated bin-score, >0.9 for all tools (cutoff >15 for Viralverify) to get a more conservative prediction for each genome. For comparison we applied the Random Forest (RF) model to the same genomes to benchmark RF performance.

Here, Viralverify achieved the highest AUC followed by the RF-model. However, Viralverify and other viral predictors scored low on F1 and MCC, which suggests that these tools are accurate on only one of the two prediction labels. Furthermore, we assessed the different approaches of summarizing the viral scores across contigs for each bin and found that the bin-score aggregation methods produced largely comparable results. In summary, across all performance metrics, we found that the RF model from the original submission of the manuscript had the best and most balanced performance in the benchmark. These results are now shown in Supplementary Figure 5 A-C and updated in the main text:

Lines 159-167: “Finally, to further validate the RF model, we compared the performance in predicting if a bin was viral or non-viral to an array of other viral predictors (**Figure 1c, Supplementary Figure 5b**). Using the mixed simulated dataset the single contig methods had much lower discriminatory performance compared to the RF model. For instance, multiple single contig viral predictors with a high AUC (up to 0.98) displayed low MCC scores meaning that the prediction was only accurate for one of the predicted labels (**Figure 1c**). In contrast, the RF-model displayed both high AUC (0.93) and MCC (0.87) as it incorporates a variety of

genome information across contigs and correctly handles both predicted labels (Supplementary Figure 5).”

Supplementary Figure 5. Evaluation of viral predictors on simulated data. A) Viral prediction performance on contigs by published viral contig predictors and B) bin-level/multi-contig-level viral evaluation including PHAMB. In (A) AUC, F1 and MCC were calculated on viral or non-viral fragmented contigs from CAMI simulated dataset with genomes from viruses and bacteria (see methods for cutoffs). In (B) prediction scores for contigs from the same genome were summarised in different ways (weighted by length, mean, median) for viral predictors and each genome was assigned viral or non-viral based on the summarised score. A prediction from the Random forest (RF) model was also assigned to each genome to calculate performance metrics for the PHAMB approach. C) ROC performance-curves shown for each method. All results here were calculated based on the simulated mixed genome dataset.

Related to the last point, how does VAMB handle plasmid sequences? Distinguishing between viruses and plasmids can be challenging, particularly if plasmids are not included in the training dataset.

We found that 16/20 Plasmids and 249/280 virus genomes were binned in the mixed genome simulated dataset, however note that only 3/16 plasmids were recovered

with high recall and precision (recall ≥ 0.9 and precision ≥ 0.95) (Supplementary Figure 3B). Based on this, it seems that plasmid binning is a more difficult task compared to binning viruses. All 16 plasmids were predicted as non-bacterial by the RF-model, so the reviewer raised a valid concern: how to distinguish between virus and plasmid? A downstream approach to distinguish plasmid-bins from *bona fide* viral bins is CheckV, which we highly recommend already as part of the standard workflow. We analysed the plasmid and viral VAMB-bins with CheckV. When summarising the annotations, we found that 8/16 of the plasmid bins were not viral-annotated by CheckV, thus described as “NA”. Furthermore, 6/16 were annotated as viral by the *de novo* HMM-based model, which is the secondary viral annotation model used for genomes displaying little overlap with the viral reference database (Supplementary Figure 6 A). On that note, we generally do not trust the HMM-based predictions as we have typically found a minimal number of viral genes and several host genes in other VAMB bins predicted as viral with this model, see Supplementary Figure 19. In comparison when we looked at the viral VAMB-bins, only a handful (13/249) were predicted using the less certain HMM-based while the remaining viral-bins were annotated viral by the AAI model. Furthermore, we found that the plasmids can be readily distinguished by their higher number of host genes as seen in Supplementary Figure 6B. In summary, the RF model cannot distinguish plasmids from viral bins, but these can be picked up downstream by asserting the viral quality statistics generated by CheckV. These results are now shown in Supplementary Figure 6 A-B.

Lines 167-171: “While the RF-model predicts plasmids incorrectly as viral, we found that the downstream use of CheckV helped making a final confident evaluation as plasmid bins contain multiple bacterial-origin genes and are typically classified as “NA” or picked up by the less precise HMM-model (**Supplementary Figure 6**). Thus, we found the RF-model to be the best suited method on bin-level in mixed-organism assembly datasets.”

Supplementary Figure 6. CheckV evaluation of binned virus and plasmids from the mixed simulated dataset. A) CheckV quality evaluation counts of plasmid and virus genomes. B) Boxplot of the proportions of viral (*viral_orf*), host (*bacterial_orf*) and unknown genes (*NA_orf*) in plasmids and virus genomes. Each distribution is separated based on the CheckV quality evaluation assigned to each bin. Results of (A) and (B) were calculated based on the simulated mixed genome dataset.

It's a great idea to compare viral genomes from single-contig assembly versus binning. However, I found some issues with the analysis in Figure 1C, in which the authors compare the number of viruses recovered with DVF (single-contig) versus

VAMB (viral-bins). It's hard to understand whether the difference is because the VAMB viral classifier is better than DVF (shown in 1B), VAMB bins are more complete (due to containing multiple contigs), or VAMB bins are larger due to contamination (since the ground truth is not known). For example, a more accurate viral prediction tool (Virsorter2 or ViralVerify) might perform considerably better than DVF and close the gap with VAMB. These issues could be addressed by using simulations instead of real data (as suggested above) and then compare the number of viral MAGs to the number of viral contigs at a minimum % completeness and % contamination level. This would directly compare the results of binning with VAMB to single contigs where the ground truth is known.

Author reply:

This is a very good suggestion and that we investigated above. Please review the results on simulated data for comparing VAMB vs the array of viral prediction tools (Supplementary Figures 5A-C).

Do the authors have a way to estimate or flag bins that contain contamination from other viruses or from bacteria? While 83% of viral bins matching the gold standard dataset were free of contamination, 17% of these WERE contaminated. An approach like viralVerify could be used to remove contigs labeled as bacteria if proteins were searched versus the Pfam database. Or contigs could be removed that contain many CheckV host markers and no CheckV viral markers or hits to VOGDB. Estimating viral contamination seems more challenging but would also be extremely helpful. For example, if a bin contained a 45 Kb circular viral contig, then any other contig in the bin is likely contaminated. Contamination in the bins will lead to overestimation of completeness and problems with data interpretation.

Author reply:

Thank you very much for the suggestion and comments on this. We realised that we did not distinguish between single-contig and multi-contig bins in this analysis, which is important for how we interpret the results. We agree with the reviewers that bin-contamination may heavily influence downstream analysis and found it imperative to expand on this part of the manuscript. In general terms of bin contamination, it's important to consider both multi-contig bins (fragmented genomes) and single-contig bins; if these represent near-complete genomes then the best case scenario is a single-contig bin. Note that it is still technically a binning challenge to keep near-complete/complete assemblies separated from the remaining contigs. However, as the reviewer correctly pointed out, it's very important to consider the magnitude of viral contamination that can lead to overestimation in fragmented genomes that are binned into multi-contig bins.

Briefly, we first identified viral bins and vOTU (viral OTU gold standard) pairs based on the best hit of each viral bin in the COPSAC dataset (at minimum ANI \geq 95% and AF $>$ 75% across the whole genome). From this we got 1,705 viral bins highly similar

to a vOTU, of which 672 were multi-contig bins. We then counted for each viral bin how many of its contigs aligned to the paired vOTU (1) and how many base pairs in total aligned (2) and how many did not (3). From this, for all bins (both single-contig and multi-contig bins) found that 91.4% of all bins contained no contaminating contigs. Now, considering only the multi-contig bins (n=672), we calculated an average bin-purity of 90.9% corresponding to an average of 9.11% contamination (example bin size 90,000bp, 8,000bp not aligned to corresponding vOTU, $(8,000/90,000)*100=8.8\%$ contamination). In addition, the median bin-purity was 100% and median contamination of 0%, so clearly we found some outlier-bins with high contamination driving the average up. We imagined that highly contaminated bins would be considerably bigger than their closest reference in the CheckV genome database. Thus, we looked up the estimated completeness by CheckV and found that 102/672 were estimated to be overcomplete as their total size exceeded their closest viral reference by >20%. Now, if we removed bins that were labeled as overcomplete by CheckV, we calculated an average contamination of 2.55% (n=570 multi-contigs bins). The table with counts and base pairs are provided in source data for Figure 3.

We then repeated this analysis based on the simulated dataset where we have a ground truth viral genome. Interestingly, we found in the simulated data that 88% of the viral bins (both single-contig n=150 and multi-contig bins n=99) had a Precision of 1, i.e. where all contigs correspond to the same genome (see new Supplementary Figure 7A). Here we found similar trends for multi-contig bins as in the real data, with an average bin-purity of 82.7% and average of 17.3% contamination but a median purity of 100% and 0% contamination. Again, if we removed bins that were labeled as overcomplete by CheckV, we calculated an average contamination of 5.5% (n=78 multi-contigs bins). From these results we would infer that the majority of multi-contig bins remain low on contamination while some outliers are highly contaminated. The table with counts and base pairs are provided in source data for simulated data for VAMB. We have added to the main text:

Lines 200-209: “Based on the viral bins (n=1,705) that were highly similar to metavirome viruses in the COPSAC dataset (see methods), we found in 91.4% of all cases, each bin contained no unrelated contigs (Figure 2d). Considering only multi-contig bins (n=570) we calculated an average bin-purity of 97.4% in base pairs (median 100%), meaning that on average 2.55% of the genome was not aligning to the corresponding MVX virus. This indicates contamination or, alternatively, a more complete virus in the bulk metagenomic dataset. We further investigated the extent of contamination based on simulated data where 87.6% of the viral bins had a precision of 1 (Supplementary Figure 7a). For multi-contig bins we calculated an average bin-purity of 94.5% (median 100%) supporting the results on real data that the majority of bins have low contamination.”

We then investigated the origin of these contaminating contigs in viral bins with

contamination. In 2/31 and 3/31 cases, the contaminants were of bacterial and plasmid origin, respectively. In the majority of cases 28/31, contaminating contigs originated from other virus genomes. However, as we outlined above the degree of contamination is overall low. We are confident that bacterial contigs with a high proportion of host-genes in viral bins are already removed by CheckV in the decontamination step. Thus, we fully agree with the reviewer that contaminating viral contigs is a much more difficult issue to resolve. One approach could be using the closest virus reference (in the CheckV database) information from CheckV for each contig and use it to discard contigs in a bin that is not in agreement with the consensus. However, we can only expect this approach to be sensible if the contigs of the same virus typically point to the same closest virus reference. We investigated how often this happens with the ground truth simulated data for the 250 virus genomes, and found that in only 27% of cases (49/193) where a simulated genome contains >1 contig there is total consensus on nearest reference, see new Supplementary Figure 7B. Hence, we do not think that there is much to gain with this approach as contigs of the same virus genome are difficult to associate using only reference based alignment processes.

Supplementary Figure 7. Contamination in simulated viral-bins and nearest reference consensus. A) Viral bin precision/degree-of-contamination for binning using VAMB on the simulated mixed genome dataset (bacteria $n=8$, plasmids $n=20$ and viruses $n=280$). Here the vast majority of bins has a Precision of 1 meaning that all contigs originate from the same genome B) The maximum nearest-reference contig consensus (in the CheckV database) within viral bins of the simulated mixed

genome dataset. A proportion of 1 indicates that all contigs in a bin match the same closest-reference genome.

Finally we agree with the reviewer that one should be on the watch for Circular contigs (complete-phages) and additional contigs in the bin should then ideally be dropped. An approach to this could be running CheckV on individual contigs in parallel and then refining bins with circular-flagged contigs by removing other contigs of that bin. However, there are some extra computational costs to this approach.

Minor comments:

Does VAMB have biases against certain types of viruses? For example, some viral genomes are quite small (e.g. <5Kb) like Microviruses, Inoviruses, CRESS-DNA viruses, RNA viruses, and others.

Author reply:

Very good question, that we now address using the simulated data. Please see the text above and Supplementary Figure 4. In conclusion, we did not find any obvious biases or issues towards small viral genomes based on the simulated dataset.

Lines 155-157: "Based on 1,340 viral bins that were highly similar to metavirome viruses in COPSAC (see methods), we found in 83.7% of all cases, every contig in the bin mapped to the virus". I assume all these bins contained at least 2 contigs? Can you confirm?

Author reply:

As we described above in the response we have redone these calculations and ensured that they were done based on bins containing at least 2 contigs. The results on the simulated data can be found in Supplementary Figure 7.

Line 243: "none of the 916 crAss-like bins could be associated with any of the 3,306 Bacteroidetes bins in our dataset, which suggests that crAss-like phages are not frequently targeted by CRISPR spacers extracted from Bacteroidetes CRISPR-Cas systems". I thought this was interesting, but I wonder how generalizable this result is. Another paper found the opposite result: <https://doi.org/10.1038/s41467-021-21350-w>. Were you able to assemble a large number of CRISPR arrays from the Bacteroidetes bins? (1) Do these crAss-like bins match CRISPR spacers from reference genomes? (2)

Thank you for this comment. We have now revisited the host-annotation of the crAss-like viruses and redone the analysis with more relaxed blastn thresholds of >95% sequence identity and >90% spacer coverage and at most 2 mismatches. Using this threshold, we found that we could annotate 74 of 916 crAss-like bins to Bacteroidetes MAGs binned in the same dataset. As suggested, we then tried to do the annotation with a larger CRISPR-database based on organisms outside the dataset (<https://academic.oup.com/nar/article/49/6/3127/6157093>), similar to what was done in the paper by Yutin *et al.* 2021. This search against a large CRISPR-spacer database revealed 512/916 host-annotations (using the same above mentioned thresholds), which is a dramatic increase from 74 where we only aligned the mined CRISPR spacers from binned MAGs to the crAss-like bins. Therefore, to answer the reviewers questions, (1) We were able to assemble CRISPR arrays (with confidently predicted subtypes) for 998/3306 (~30%) of the Bacteroidetes bins, for this we used the same tools as Yutin *et al.* (2) We downloaded the CRISPRopen database based on 580,383 bacterial genomes and aligned the spacers from these to crAss-like viruses. By filtering hits with the blast-hit cutoffs as when we mapped to MAGs, we were able to host-annotate more crAss-like bins. We do think that this result still shows that crAss-like viruses are less frequently host-annotated by bacteria in the same environment. We have added to the main text:

Lines 295-304: “Interestingly, because the host range of crAss phages are not well understood we investigated CRISPR spacer hits to the MAGs in our databases. Even though we could host-annotate an overall of 45.3% of all HQ viral populations to a MAG, only 74 of the 916 crAss-like bins could be associated with any of the 3,306 Bacteroidetes bins in our dataset using CRISPR spacers. This was despite having assembled CRISPR arrays (with confidently predicted subtypes) for 998/3306 (~30%) of the Bacteroidetes bins. When we performed a similar search to a comprehensive CRISPR spacer database (Dion *et al.* 2021) of 580,383 bacterial genomes we could annotate 512 of the 916 crAss-like bins to Bacteroidetes bacteria. These findings suggest that crAss-like phages are not frequently targeted by CRISPR spacers extracted from Bacteroidetes CRISPR-Cas systems within the same environment.”

Did binning allow you to recover genomes of any very large viruses? What was the size of the largest viral bins, and are they likely to represent pure viral genomes?

Author reply:

We appreciate the reviewers interest in this part of the paper and we lightly addressed it in now Supplementary Figure 8. Here we counted the number of viral bins with a total size of >200,000 bp (aka. jumbo viruses) in the HMP2 dataset. In light of your comments, we found it interesting to revisit and address the concept of viral purity in these putative jumbo viruses. Before tallying up the results from CheckV for Supplementary Figure 8, we did filtering based on the quality and completeness statistics from CheckV. Basically, we removed jumbo bins predicted

with the HMM-marker-model and kept only those with a high AAI to a known viral-family (the AAI-based model) in the CheckV database. The filtered bins displayed a median of 0% contamination (max contamination 10.4%) according to CheckV. However, the border between viral gene and bacterial gene content is blurry for Jumbo viruses as these viruses encode enzymes also commonly found in bacteria and archaea (<https://www.nature.com/articles/s41467-020-15507-2>). Therefore, strictly enforcing 0% bacterial contamination might not be the right strategy for this type of virus.

If we allow up to 10% of all genes to be annotated as host genes, we achieve a list of bins in the size-range 200,290 - 402,087bp (n=54) across the Diabimmune, COPSAC and HMP2 datasets. We believe it's worth noting that 25/54 bins are "overcomplete", which means their genome size is 20% bigger than their closest viral family in the CheckV database. However, these bins are characterised by a low proportion of host-genes and relatively high number of viral genes. We believe these bins are likely jumbo viruses but further validation is required for absolute certainty. We have added a table with quality-statistics for these Jumbo bins as Supplementary Data 3 and updated the following in the main text:

Lines 223-226: "We also observed an increase in genome completeness for larger viruses/jumbo viruses with a genome size > 200 kbp compared to single contig evaluation (**Supplementary Figure 8**). Across all the datasets we observed 54 binned putative jumbo viruses (**Supplementary Data 1**)."

Line 263: what is the significant of the reverse transcriptase genes?

Author reply:

The reverse transcriptase (RT) genes are known components of diversity generating regions in bacteriophages (<https://www.nature.com/articles/s41467-021-23402-7>). Many of these RT genes can be annotated taxonomically according to host specificity. We highlighted these RT genes as they were frequently genomically annotated and could be relevant for further downstream analysis. This observation had not been highlighted before the above mentioned publication. We updated the main text:

Lines 322-326: "Finally, Reverse Transcriptase (RT, PF00078) proteins were also highly detected, in agreement with recent results and shared by all viral populations irrespective of the predicted host (**Supplementary Figure 17A**). These proteins are known modules in bacteriophage diversity generating regions that cause hypervariability in specific viral genes (Benler et al. 2018)"

The authors identify "hotspots for viruses". Might these patterns be partially explained by (i) the prevalence of the CRISPR-cas system, and the (ii) total number of MAGs from these groups?

Author reply:

This is a very good observation. The total number of MAGs and the sequencing depth of these may partially explain this and we see for some bacteria that the many viral-associations are also made through the prophage signature so not necessarily only CRISPR-spacers. We think this could be a relevant point in the discussion, i.e. what factors that influence our ability to establish virus-host connections. We now discuss this:

Lines 363-371: "Several of these genera represent not only highly abundant gut commensals but also hotspots for viruses as we have shown by connecting 230 and 123 viral populations to *Bacteroides* and *Faecalibacterium*, respectively. Viral hotspots could be partially explained by factors such as their absolute numbers and genome sequencing depth, which may allow for a more complete assembly of CRISPR-cas systems. A large part of these connections were also made via prophage signatures, i.e. shared genomic elements between bacteria and phage (Figure 5). Prophage signatures could be the result of increased rates of lysogeny and coinfection as higher microbial densities and phage adsorption rates provide favorable conditions for multiple phages to "piggyback" highly productive hosts and exchange genetic material (Luque and Silveira 2020)."

Line 284: The authors conclude that "evidently, a significant portion of the sequenced microbiome remains dark-matter while HQ viruses identified in this study only accounts for a small fraction of the sequenced space.". I feel this conclusion was not adequately supported from the data. Dark matter bins (not matching and reference virus) could result from a number of technical artifacts, including (i) short incomplete bins, (ii) contaminated bins, (iii) non-viral bins. I believe the CheckV database includes a very large number of human gut viruses and has very good coverage of the gut virome, so I suspect these technical issues might be having an outsized impact.

We wholly agree with the reviewer that Dark matter bins might represent both incomplete, contaminated and non-viral bins. We do provide evidence that we can identify some viral-like bins with high viral prediction scores and targeted by CRISPR-spacers (could be incomplete viral bins), but those we do separate from the dark-matter ones in the text. We think we are on the same page and added the following sentence to highlight the three points of the reviewer.

Lines 345-348: "However, caution should be taken with labelling dark-matter bins as viruses since these are possibly incomplete, contaminated or contain other types of

mobile genetic elements that encode proteins shared with viruses such as integrases, polymerases and toxin-antitoxin addiction modules (Mruk and Kobayashi 2013; Makarova et al. 2009).”

Reviewer #2 (Remarks to the Author):

The authors describe a method and its application for thorough analysis of the virome from metagenomic samples -- both those enriched for viruses and those directly sequencing. This manuscript is well done. It is clearly explained and contains adequate detail. The approach is noteworthy as other methods classify individual contigs to classify but this includes a binning step and training of a RF model to classify the bins.

The authors point to GitHub accounts that contain their code and workflow. My one critique is that the GitHub account or the workflow is incomplete (some steps are noted as "in progress"). Ideally this would be complete, as it would provide assistance to those who wish to use the method that the authors developed.

Thank you very much for the suggestion, we fully agree that the GitHub page should be complete and we have now updated it with more information that allows users to run the PHAMB method on binned metagenomic contigs. We have removed a lot of old code, simplified the scripts for gathering annotation and running the Random Forest model based on VAMB clusters.

https://github.com/RasmussenLab/phamb/tree/master/workflows/mag_annotation

We have also provided a directory named vCAMISIM with the script and functions to rerun our benchmark on Dataset A and to inspect the code used. It contains the pooled gsa-file from CAMISIM, vamb-clusters and viral annotation files for all of the predictors benchmarked on the pooled simulated assembly. Finally, source files used to make the plots for the simulation results are also there and can be readily recreated with the right python dependencies installed (most importantly scikit-learn v. 0.21.3). The code provided is highly generalisable for other use cases as well, i.e. using metabat clusters as input instead of vamb clusters or running CAMISIM on a different set of genomes.

<https://github.com/RasmussenLab/phamb/tree/master/vCAMISIM>

We also decided that it was out of the scope of this project to provide complete pipelines for automating phage-host annotation and viral abundance since (1) many different approaches can be taken to this and (2) dedicated tools for this already exist for this i.e. CRISPROpenDB (<https://github.com/edzuf/CrisprOpenDB>) or Wish (<https://github.com/soedinglab/WISH>) for host-annotation and CoverM (<https://github.com/wwood/CoverM>) for viral abundance.

REVIEWER COMMENTS

Reviewer #1 (Remarks to the Author):

In my view, figure 1 is the most important figure in the paper, as it validates the accuracy of PHAMB relative to other tools. Unfortunately, I find that the analyses present here are not executed in a consistent way, not clearly described, and hard to follow. It should be very easy for readers to follow these experiments and to present the data in a way that is not biased towards PHAMB. Here are a few specific comments on figure 1:

-The authors write that "Bins from any metagenome can be parsed through the RF model". My understanding was that the RF model could only be applied to VAMB bins. Please indicate if these must be VAMB bins or not.

-Are 1b and 1c based on the same data? Or different data? Please clearly indicate the datasets evaluated in 1b and 1c in the figure legend, and whether the input bins are the same to all tools (i.e. VAMB bins or bins from another tool).

-The same viral prediction methods and performance metrics should be shown in 1b and 1c, assuming these are two different datasets. Currently only DVF is evaluated in 1b while the performance metric changed between 1b (ROC AUC) and 1c (MCC).

-For 1c, the score cutoff used by the authors (0.75 and 5 for viralverify) seems somewhat arbitrary, while I presume the score cutoff used by PHAMB was machine optimized. A poorly chosen score cutoff will inflate the relative performance of PHAMB. Supporting this point is the data shown in Supplementary figure 5, where the AUC values for various tools (cutoff independent) is quite good and in some cases exceeding PHAMB (e.g. viralverify). It is only when the cutoffs are applied that PHAMB is clearly superior. I would suggest only showing the AUC statistic, which is score independent. Otherwise, the authors will need to show the maximum performance for statistics (F1, MCC) for each tool.

-For 1d (application to real data), please include the other viral predictions methods (shown in 1c) for evaluation of the single contig method. Also indicate the score threshold used for the individual tools. I'm concerned that the relative performance of PHAMB is being inflated by only showing comparison to one tool (DVF).

-As an aside (optional), for 1d, I'm also interested to know (supplement or text) the % of high-quality single-contig viral genomes identified using the existing approaches (e.g. VirSorter2) that are not found in any high-quality PHAMB bin. And whether these viral contigs missed by PHAMB share any common properties. It's important to know whether PHAMB has any blind spots compared to existing approaches.

Reviewer #2 (Remarks to the Author):

The authors have adequately addressed the reviewers' comments.

Once more we would like to thank the reviewers for their time and efforts in reviewing our manuscript. We appreciate the comments and suggestions, and have made efforts to improve benchmark consistency and clarity of the results hereof. In particular we have streamlined benchmarking of the CAMI simulated genomes and the real data featured in Figure 1. Furthermore, we now feature all performance metrics for both benchmarks and extensive source data for this is provided.

Remarks to authors (only Reviewer 1)

The authors write that "Bins from any metagenome can be parsed through the RF model". My understanding was that the RF model could only be applied to VAMB bins. Please indicate if these must be VAMB bins or not.

We acknowledge that we were not precisely clear in the particular sentence. Therefore, we have now specified that bins produced by VAMB from any metagenome such as marine, human-gut or soil can be parsed through the RF model. Furthermore, as the Random Forest model has been trained on bins produced by VAMB it will be possible to use it on bins produced by other binners. Although we have not benchmarked this, it is therefore possible to use PHAMB on output on any metagenomics dataset and bins from any binner as input.

Lines 854-858 in Figure 1 legend: Viral and bacterial labelled bins were used as input for training and evaluating the RF model. Bins from any metagenome such as human gut, soil or marine can be parsed through the RF model to extract a space of putative viral bins that are further validated for HQ viruses using dedicated tools like CheckV.

Are 1b and 1c based on the same data? Or different data? Please clearly indicate the datasets evaluated in 1b and 1c in the figure legend, and whether the input bins are the same to all tools (i.e. VAMB bins or bins from another tool).

We appreciate this catch, we had not indicated which datasets were used in 1b and 1c in the figure text and only described it in methods - this is now indicated more clearly in the Figure and in the legend. The data used as input for 1b was the Diabimmune dataset and the input for 1c was from the combined dataset of the viral-simulation analysis.

>>We have changed Figure 1c to Figure 1e, from here on we refer to the new figure label.<<

The rationale for Figure 1b and Figure 1e:

For Figure 1b we wanted to test how good VAMB+PHAMB was at identifying and reconstructing viral genomes from the bulk metagenomics dataset (MGX). Viral bins in the MGX data were identified as described in methods using the gold standard from the paired metaviromic dataset (MVX). The RF model was trained on the COPSAC dataset and for Figure 1b we generated AUC curves by evaluating performance on the Diabimmune dataset. We had only included DVF in the figure because we use DVF as input til the RF model as well and because it was shown in (**Supplementary Figure 5a**) to have the best single-contig prediction performance. We now include all of the methods in **Figure 1b**.

For Figure 1e we used the mixed CAMI simulated dataset (bacteria, phages and plasmids) as input. Here we did not use any bin information, but rather the ground truth for evaluation.

We have thus updated the legend for Figure 1:

Lines 860-867 in Figure 1 legend: b) AUC, F1-score and Matthews correlation were calculated for prediction results on viral bins from Diabimmune. These performance scores were calculated based on probability scores from the trained RF model and summarised viral-bin scores of various viral prediction tools. For all tools except the RF model, genomes were labelled viral if the summarised viral-score across all contigs, calculated either as a mean, median or contig length weighted mean passed a threshold. The following thresholds used were 7, 0.5, 0.9, 0.9, 0.9 for Viralverify, Seeker, Virsorter2, Virfinder and DeepVirfinder, respectively.

Lines 874-876 in Figure 1 legend: e) Similar to (b) prediction performance scores were calculated for the trained RF model and various viral predictors but on prediction results of CAMI simulated viral genomes from the mixed genome set including bacteria, viruses and plasmids.

The same viral prediction methods and performance metrics should be shown in 1b and 1c, assuming these are two different datasets. Currently only DVF is evaluated in 1b while the performance metric changed between 1b (ROC AUC) and 1c (MCC).

We have included all of the viral predictors we used in Figure 1e in the Figure 1b performance test on viral bin prediction from real data. In addition, we now report both the AUC, F1 and MCC for Figure 1b and Figure 1e. Furthermore, we use the same thresholds for the viral predictors, these are indicated further down in the response.

Lines 123-128 in main text: Here, we found that the RF model was able to separate bacterial and viral clusters very effectively with an Area Under the Curve (AUC) of 0.99 and a Matthews Correlation Coefficient (MCC) of 0.91 on the validation set (Figure 1b and Supplementary Table 1). Compared to single-contig-evaluation methods, the RF model was superior as other methods achieved an AUC of up to 0.86 and MCC up to 0.16

For 1c, the score cutoff used by the authors (0.75 and 5 for viralverify) seems somewhat arbitrary, while I presume the score cutoff used by PHAMB was machine optimized. A poorly chosen score cutoff will inflate the relative performance of PHAMB. Supporting this point is the data shown in Supplementary figure 5, where the AUC values for various tools (cutoff independent) is quite good and in some cases exceeding PHAMB (e.g. viralverify). It is only when the cutoffs are applied that PHAMB is clearly superior. I would suggest only showing the AUC statistic, which is score independent. Otherwise, the authors will need to show the maximum performance for statistics (F1, MCC) for each tool.

You are correct that the score cutoffs for these tools were chosen arbitrarily and conservatively based on the documentation from the tools individual githubs. We revisited the githubs of all tools and found more suggested cutoffs that we have applied now (see below). There are no clearly described cutoffs for DeepVirFinder and Virfinder so we applied a conservative score cutoff of 0.9 for both tools.

From Github:

Viralverify: 7 (found on github)

Seeker: 0.5 (found on github)
Virsorter2: 0.9 (found on github)
DeepVirFinder: 0.9 (no information)
Virfinder: 0.9 (no information)

As there is no precedence on how to aggregate these scores on genome-level, we used the same cutoffs as stated above for when the contig scores are summarised on genome level. We found no drastic changes overall in terms of performance of the various tools using these new cutoffs.

However, as the Reviewer correctly points out, the score threshold for PHAMB to predict from a set of contigs (a bin) was machine optimized during training of the method. This has of course not been done for any of the other methods as they were designed to identify single contigs and not sets of contigs (bins). We therefore, to make a fair comparison, optimized the thresholds for each method based on the test data (Diabimmune). When we then evaluate on the same data, this will be overfitting, however it will also provide us with an indication of near maximum (overfitted) performance of the single-contig methods that we compare against.

To do this we investigated the prediction score distributions (in this plot the mean score) for each tool and coloured distributions according to the truth genome-label. For every tool (except PHAMB that is machine optimised) we have indicated with a solid black line the Github-based threshold and with a dashed line the optimized (overfitted) threshold that we used for each tool that determines if a genome is *bacterial* or *viral* based on the summarised genome score (**Supplementary Figure 5b**).

Optimized thresholds:

Viralverify: -1.3
Seeker: 0.75
Virsorter2: 0.9
DeepVirFinder: 0.5
Virfinder: 0.5

On MCC and AUC: We note that we find it important to include the MCC in the figure as a user of any of these methods will have to set a threshold to define if a bin is viral or not. However, we also agree that a threshold-free approach such as the AUC is very informative and we therefore now include that as well for the analyses. Additionally, we include the confusion matrices (based on the initial and optimised thresholds) in (**Supplementary Figure 6**).

Furthermore, we agree with the Reviewer that it was curious for Viralverify to perform so well based on AUC but quite poorly in terms of F1 and MCC in the viral simulation experiment. Conversely, we found it curious that PHAMB performed so well overall but did not reach as high an AUC as Viralverify. Based on the score distributions below (**Supplementary Figure 5b**)

Supplementary Figure 5b. Density plots of summarised genome scores for simulated CAMI genomes. The summarised genome scores are displayed for each virus prediction tool (length weighted mean) and PHAMB (probability score) then colored by the ground truth genome label.

Using the new cutoffs indicated by the dashed-black line we observed a great improvement in MCC for all tools except for PHAMB and Virsorter2 where a new cutoff was not applied. For i.e. Viralverify, the MCC increased from 0.1 to 0.39. As the MCC is an aggregate score of the confusion matrix, the improvement is easily observed by looking into the confusion matrix for the various tools based on the initial cutoff and the overfitted/ideal cutoff (**Supplementary Figure 6**). Nevertheless, the RF-model still shows the best MCC score even after overfitting the other tools to the dataset (**Supplementary Figure 6**).

Supplementary Figure 6. Confusion matrices of predictions for simulated CAMI genomes. First row (A) shows the confusion matrices of each virus prediction tool based on an initial cutoff and second row (B) shows the confusion matrices based on ideal cutoffs on the data. In (C) the confusion matrix for PHAMB is displayed.

While the MCC and F1 scores are dependent on the applied cutoff and represent a snapshot of a tool's performance, the AUC is cutoff independent. Viralverify does achieve the highest AUC in the simulation experiment and albeit lower but still high in the real data-set evaluation in Figure 1e. However, looking at the mean-score distributions (**Supplementary Figure 5b**), there is a score-overlap between bacterial and viral genomes that ultimately results in a lower MCC score than the RF model. Interestingly, we found one misclassified bacterial genome by the RF model (with a high prediction score) that punished PHAMB's AUC score to 0.93 despite the very high MCC score of 0.87. In summary, we do not believe the RF-model performance is inflated on this dataset and we note and appreciate that Viralverify is out of the box a great alternative candidate for identifying viral-like bins based on these summarised genome-scores.

Lines 164-174 in main text: For instance, multiple single contig viral predictors with a high AUC (up to 0.98) displayed low MCC scores meaning that the prediction was not very accurate at the given threshold (Figure 1e, Supplementary Figure 5-6). We then tried to optimize the decision threshold for each of the single-contig viral predictors (Supplementary Figure 5-6) which improved the MCC slightly. For instance, Viralverify achieved an AUC of 0.98 on the simulated data, showing that it was effective in separating bacterial and viral genomes, however with an overlap in the bacterial and viral score distributions. Therefore, even with an optimized threshold, Viralverify displayed an MCC of 0.39. In contrast, the RF-model displayed both high AUC (0.93) and MCC (0.87). Thus, we found the RF-model,

followed by Viralverify, to be the best suited method on bin-level in mixed-organism assembly datasets.

For 1d (application to real data), please include the other viral predictions methods (shown in 1c) for evaluation of the single contig method. Also indicate the score threshold used for the individual tools. I'm concerned that the relative performance of PHAMB is being inflated by only showing comparison to one tool (DVF).

In Figure 1d, we compare the number of identified viruses by their estimated completeness calculated using CheckV within a subset of contigs in each dataset. To extract this subset, we applied the RF-model that confidently distinguishes larger bacterial bins from putative viral bins. In short, the finite subset of contigs does not contain contigs binned into larger bacterial bins. Now, within this subset of contigs, the best-case scenario number of viruses that can be identified on a single-contig-level with a given degree of completeness (Complete, HQ or Medium-quality) is displayed by the yellow bar in Figure 1d. If we run a series of viral predictors (i.e. Virsorter2) the number of HQ-contigs predicted by Virsorter2 can never exceed the yellow bar, as it will always be a subset of the yellow bar. What we illustrate in Figure 1d and the main point of the figure is: when we bin some of these contigs by cluster-information from VAMB binning and parse them through CheckV, we observe a great improvement in the total number of viruses identified with a given degree of completeness. Evidently, as we have also shown via viral genome simulation experiments, we can bin viral contigs, which may correspond to viruses of lower completeness evaluated on a single-contig level on their own, and instead recall more complete viruses this way. Therefore, with the point of this figure in mind, we find it out of scope to add additional bars for other viral predictor tools here.

As an aside (optional), for 1d, I'm also interested to know (supplement or text) the % of high-quality single-contig viral genomes identified using the existing approaches (e.g. VirSorter2) that are not found in any high-quality PHAMB bin. And whether these viral contigs missed by PHAMB share any common properties. It's important to know whether PHAMB has any blind spots compared to existing approaches.

With the response to the Figure 1d request above in mind, we investigated and found that 97.7%, 95.9% and 95.3% of HQ contigs can be found in a HQ bin in the COPSAC, Diabimmune and HMP2 dataset, respectively. This means that up to 2.3%, 4.1% and 4.7% of the HQ contigs are not found in a HQ PHAMB bin, no matter what tool makes the prediction. This makes sense because another tool can never predict more HQ contigs than what we established with CheckV already. Nevertheless, this blind spot can be rescued by running CheckV in parallel on the individual contigs and identify HQ contigs not found in a HQ bin. We think the "common properties" research question is interesting but probably a time-demanding and technical investigation that we deem more relevant for a future study.

Lines 132-136 in main text: Based on the single-contig CheckV evaluations, we found that 97.7%, and 95.3% of HQ contigs were binned into HQ bins in COPSAC and Diabimmune, respectively. This means that a small percentage of the HQ contigs, up to 2.3% and 4.7%, are lost in the binning process at the expense of a net increase in genome recovery but can be recovered by parallel single-contig evaluations.

REVIEWERS' COMMENTS

Reviewer #1 (Remarks to the Author):

I thank the authors for addressing all of my technical questions. I have just a few final comments:

During the review of the manuscript two new papers have published on the same topic:
<https://doi.org/10.1093/bioinformatics/btab213> and
<https://www.biorxiv.org/content/10.1101/2021.12.16.473018v1>. I'd encourage the authors to cite this work in their discussion (at least the non-preprint one).

Figure 1b: the low relative performance of virsorter2 is surprising. I'd encourage the authors to double check this is accurate.

Line 510: The authors seem to use 2 different definitions of a Complete genome, one based on 100% estimated completeness and the other based on terminal repeats (as reported by CheckV). Figure 1c shows that PHAMB yields a greater number of complete genomes, but this would only be consistent with the first definition of a complete genome. In the methods, the authors write: "the quality of each MVX virus we organised the success of genome recovery into the 4 CheckV levels (Low-quality $\leq 50\%$, Medium-quality $\geq 50\%$, High-quality $\geq 90\%$, Complete =100%). Complete = Closed genomes based on direct terminal repeats (DTR), inverted terminal repeats (ITR)"

Line 877: "The increase..." text should be associated with panel d not panel e.

Remarks to authors

During the review of the manuscript two new papers have published on the same topic: <https://doi.org/10.1093/bioinformatics/btab213> and <https://www.biorxiv.org/content/10.1101/2021.12.16.473018v1>. I'd encourage the authors to cite this work in their discussion (at least the non-preprint one).

We thank the Reviewer for putting these two papers to our attention. We agree that the published paper, essentially a novel binning framework dedicated to and trained on virus genomes, should be mentioned in the discussion paragraph. We ponder how this binner would perform on bulk metagenomics dataset as it is only benchmarked on 3 metavirome samples (not containing larger organisms like bacteria), where it clearly outperforms MetaBAT2 as seen in their Figure 6. Depending on the tool's usage, it is crucial for a binner to also perform well in the presence of many non-viral contigs i.e. if applied to bulk metagenomics. Nevertheless, we appreciate and applaud the mutual interest in viral binning to improve studies in this biological realm.

We have added the following regarding the published publication:

Lines 393-401 in discussion: “Metavirome studies have until now been the primary source for exploring viral diversity in microbiomes. Now, viral populations are increasingly uncovered in bulk metagenomes and we showed that more complete viral genomes can be identified via viral binning across three different cohorts, similar results were found in a recent paper focused on binning of sequenced viral particles (**Arisdakessian et al., CoCoNet: an efficient deep learning tool for viral metagenome binning, 2021**). Our approach allowed precise clustering of both viral and bacterial populations in three cohorts that enabled direct investigation into viral-host interactions and discovery of new diversity. We believe that future studies can greatly leverage this approach to conduct virome analyses and investigate the viral influence of the intricate microbiome ecosystem that governs human health.”

Figure 1b: the low relative performance of *virsorter2* is surprising. I'd encourage the authors to double check this is accurate.

We also found this result interesting as *Virsorter2* performs very well on the CAMI simulated genomes. We double checked the contig annotation files and confirmed that *Virsorter2* successfully processed all contig files for the Diabimmune data. However, we note that *Virsorter2* did not produce a prediction for every contig. *Virsorter2* produced a prediction for 25% and 34% of contigs in the Diabimmune dataset and the CAMI genome contigs, respectively. Simultaneously, other tools such as *Virfinder* or *Viralverify* produced a prediction for every contig. Thus, we think that *Virsorter2*'s prediction performance may be hindered by its general ability to annotate an extensive set of contigs as those found in bulk metagenomics.

Line 510: The authors seem to use 2 different definitions of a Complete genome, one based on 100% estimated completeness and the other based on terminal repeats (as reported by CheckV). Figure 1c shows that PHAMB yields a greater number of complete genomes, but this would only be consistent with the first definition of a complete genome. In the methods, the authors write: "the quality of each MVX virus we organised the success of genome recovery into the 4 CheckV levels (Low-quality $\leq 50\%$, Medium-quality $\geq 50\%$, High-quality $\geq 90\%$, Complete =100%). Complete = Closed genomes based on direct terminal repeats (DTR), inverted terminal repeats (ITR)"

We acknowledge the confusion introduced by this paragraph and we can assure the Reviewer that we have stuck to only one definition of a Complete genome in this work. We only describe a bin or genome as Complete if DTRs or ITRs are present, as determined with CheckV. We have fixed the paragraph in the methods section to the following to avoid any confusion.

Lines 510-514 in methods: "By labeling the quality of each MVX virus we organised the success of genome recovery into the 4 CheckV levels (Low-quality $\leq 50\%$, Medium-quality $\geq 50\%$, High-quality $\geq 90\%$, Complete = Closed genomes based on direct terminal repeats (DTR) or inverted terminal repeats)."

Line 877: "The increase..." text should be associated with panel d not panel e. That is absolutely correct. We have now implemented the correction in the main figure text.